# Zinc anode-compatible in-situ solid electrolyte interphase via cation solvation modulation

Huayu Qiu[1,2,4], Xiaofan Du[1,4], Jingwen Zhao[1]*, Yantao Wang[1], Jiangwei Ju[1], Zheng Chen[1], Zhenglin Hu[1], Dongpeng Yan ⬤ [3], Xinhong Zhou[2]* & Guanglei Cui[1]*

The surface chemistry of solid electrolyte interphase is one of the critical factors that govern the cycling life of rechargeable batteries. However, this chemistry is less explored for zinc anodes, owing to their relatively high redox potential and limited choices in electrolyte. Here, we report the observation of a zinc fluoride-rich organic/inorganic hybrid solid electrolyte interphase on zinc anode, based on an acetamide-Zn(TFSI)$_2$ eutectic electrolyte. A combination of experimental and modeling investigations reveals that the presence of anion-complexing zinc species with markedly lowered decomposition energies contributes to the in situ formation of an interphase. The as-protected anode enables reversible (~100% Coulombic efficiency) and dendrite-free zinc plating/stripping even at high areal capacities (>2.5 mAh cm$^{-2}$), endowed by the fast ion migration coupled with high mechanical strength of the protective interphase. With this interphasial design the assembled zinc batteries exhibit excellent cycling stability with negligible capacity loss at both low and high rates.

[1] Qingdao Industrial Energy Storage Research Institute, Qingdao Institute of Bioenergy and Bioprocess Technology, Chinese Academy of Sciences, Qingdao 266101, P. R. China. [2] College of Chemistry and Molecular Engineering, Qingdao University of Science and Technology, Qingdao 266042, P. R. China. [3] College of Chemistry, Beijing Normal University, Beijing Key Laboratory of Energy Conversion and Storage Materials, Beijing 100875, P. R. China. [4]These authors contributed equally: Huayu Qiu, Xiaofan Du. *email: zhaojw@qibebt.ac.cn; zhouxinhong@qust.edu.cn; cuigl@qibebt.ac.cn

Multivalent-ion ($Mg^{2+}$, $Zn^{2+}$, $Ca^{2+}$, etc.) batteries (MIBs) are highly desirable for large-scale stationary energy-storage systems because of their abundant reservoir, environmental friendliness, intrinsic safety and comparable or even superior capacities to those of Li-ion counterparts[1–5]. Among the anode materials developed for MIBs, metallic Zn offers a better insensitivity in oxygen and humid atmosphere[6,7], which broadens the availability of electrolytes and lowers the handling and processing costs. Additional enthusiasm for the Zn chemistry is stimulated by its high volumetric capacity ($5855\ Ah\ L^{-1}$), superior to Li ($2061\ Ah\ L^{-1}$), Ca ($2072\ Ah\ L^{-1}$), and Mg ($3833\ Ah\ L^{-1}$) counterparts[8]. Indeed, since the "rediscovery" of the rechargeable Zn-ion batteries (ZIBs), new cathode materials and Zn-storage mechanisms have enjoyed substantial achievements in the last few years[8,9]. However, there is a fly in the ointment: the suboptimal cycling efficiency resulting from uncontrolled dendrites and notorious side-reactions occurred at the Zn-electrolyte interface (especially for aqueous electrolytes) restricts the development of real rechargeable ZIBs and their broad applicability[6,10,11].

Actually, intensive previous investigations have been dedicated to handling these Zn-related issues, such as introducing additives into electrolytes or electrodes, constructing nanoscale interface and designing hierarchical structures[12–15], but still suffer from a low Coulombic efficiency (CE). Very recently, highly concentrated electrolytes were introduced to stabilize the Zn anode by regulating the solvation sheath of the divalent cation[16–18], which is a feasible approach to reducing water-induced side-reactions and improving Zn plating/stripping CE. Unfortunately, the underlying mechanism on the inhibition of Zn dendrites has not been entirely understood yet. Even today, the Zn-electrolyte interface instability remains challenging, and a broadly applicable interfacial protection strategy is highly desired yet largely unexplored, especially compared with the rapid progress regarding the effective utilization for alkaline metal (Li or Na, etc.) anodes.

When encountered with a similar dilemma of intrinsic limitations on anodes, Li-ion batteries (LIBs) offer a tactful response: in situ formation of a solid electrolyte interphase (SEI). Admittedly, this SEI is highly permeable for Li ions and prevents excess Li consumption by blocking solvents and electrons[19,20]. Importantly, via electrolyte modulation (e.g., introduction of F-rich species), additional unusual functionalities can also be achieved, in particular concerning the dendrite suppression and long-term cycling for Li-metal batteries at high CE or high rate[20,21]. However, such SEI response has always been associated with aprotic electrolytes. Given the much higher redox potential of $Zn/Zn^{2+}$ couple ($-0.76\ V$ vs. NHE) compared with that of $Li/Li^{+}$ ($-3.04\ V$ vs. NHE), routine anions and organic solvents are difficult to decompose reductively before Zn deposition. Despite the great appeal for aqueous Zn anodes, the competitive $H_2$ evolution reaction inevitably occurred during each recharging cycle makes this in situ protection mechanism infeasible[22], while the local pH change induces the formation of ionically insulating byproducts which has been also faced in other multivalent metal anodes[23,24]. Hence, another possibility for designing reliable Zn-anode SEI is to look beyond the conventional water-based and organic-solvent electrolytes.

We explore the in situ formation of a $ZnF_2$-rich, ionically permeable SEI layer to stabilize Zn electrochemistry, by manipulating the electrolyte decomposition based on a eutectic liquid with peculiar complexing ionic speciation. Regulating the solvation structure (either locally or totally) has been verified and found to be an effective strategy for shifting the reductive potentials of electrolyte components[20,21,25]; however, due to the high charge density of $Zn^{2+}$, Zn salts do not readily dissociate in common solvents over a wide concentration range, resulting in

limited control over the coordination properties. As a new class of versatile fluid materials, the deep eutectic solvents (DESs), generally created from eutectic mixtures of Lewis or Brønsted acids and bases that can associate with each other, have been found to be interesting on account of their excellent dissolution ability, even for the multivalent metal salts and oxides[26,27]. Remarkably, characterized by highly adjustable compositions and rich intermolecular forces, DESs are also expected to accommodate concentrated ionic species and have aided the development of alternative media for electrochemistry[27,28].

Here in this work, based on a new DES composed of acetamide (Ace) and $Zn(TFSI)_2$, a large portion of $TFSI^-$ is found to coordinate to $Zn^{2+}$ directly in the form of anion-containing Zn complexes ($[ZnTFSI_m(Ace)_n]^{(2-m)+}$, $m = 1$–2, $n = 1$–3), which induces the preferential reductive decomposition of $TFSI^-$ prior to Zn deposition. Correspondingly, a well-defined anion-derived SEI layer compositionally featured with a rich content of mechanically rigid $ZnF_2$ and $Zn^{2+}$-permeable organic (S and N) components can be obtained during the initial cycling. This SEI-coated Zn anode is stabilized to sustain long-term cycling (>2000 cycles; average Zn plating/stripping CE of 99.7%), and enables a highly uniform Zn deposition even at a high areal capacity of 5 $mAh\ cm^{-2}$, without short-circuit or surface passivation. Moreover, the transformed interfacial chemistry has been further confirmed by the unprecedented reversibility of Zn redox reactions upon implanting the SEI-coated Zn anodes into cells with routine aqueous electrolytes. With this in situ anode protection, ZIBs paired with a $V_2O_5$ cathode accomplish the cyclability of 92.8% capacity retention over 800 cycles (99.9% CEs after activation), and are demonstrated to cycle up to 600 times along with a capacity fading of only 0.0035% $cycle^{-1}$ under a practical cathode-anode coupling configuration ($Zn:V_2O_5$ mass ratio of 1:1; areal capacity of >0.7 $mAh\ cm^{-2}$). As per our knowledge, it is the first successful attempt to in situ construct reliable SEI on Zn anode, providing fresh insights for all multivalent chemistries confronted with the same requirements at anodes.

## Results

**The new $Zn(TFSI)_2$/Ace eutectic solution and physicochemical properties.** Recent progress on new electrolytes has demonstrated that better control over the metal coordination environment provides more possibilities for achieving unique properties beyond routine views, such as significantly extended electrochemical window, enhanced oxidative/reductive stability and unusual ion-transport behavior[29]. In fact, there have been reports on $ZnCl_2$-based DESs, in which various complex anionic (e.g., $[ZnCl_3]^-$, $[ZnCl_4]^{2-}$, and $[Zn_3Cl_7]^-$) and cationic (e.g., $[ZnCl(HBD)_n]^+$; HBD, hydrogen bond donor) $Zn^{2+}$ species can be detected[30,31]. The Cl-containing solutions, however, are corrosive to common battery components especially at high operation voltages, and are not readily available in forming a stable and ionically conducting interphase for Zn anodes. As an alternative anion to form DESs, $TFSI^-$ is generally considered to be a decomposition source to promote the formation of uniform SEI and thus has aroused our concern[12,32]. Besides, the binding energy of $TFSI^-$ to metal ions appears to be relatively lower compared with those of other conventional anions such as $BF_4^-$, $PF_6^-$, allowing the TFSI-based salts to dissociate easily[33,34]. Based on the above considerations, the $Zn(TFSI)_2$-based eutectic solvent (ZES) with Ace as the HBD was selected as a promising Zn electrolyte.

As shown in Supplementary Fig. 1, the ZESs are homogenous and transparent liquids at ambient-temperature when $Zn(TFSI)_2$ and Ace were blended in a predetermined molar ratio range (1:9 −1:4). Correspondingly, they are denoted as ZES 1:$x$ solutions

($x = 4$, 5, 7, and 9). The lowest-eutectic temperature of ZES is found to be −51.51 °C at a molar ratio of 1:9, and rises to −35.70 °C with the molar ratio increased to 1:4 (Supplementary Fig. 2 and Supplementary Table 2). Furthermore, no phase change is observed in all ratios blow 100 °C, and weight losses are only about 4.3% (1:9) and 3.3% (1:4) after heating at 100 °C (Supplementary Figs. 2, 3), reflecting the thermal adaptability of ZESs in the operating temperature region. This wide temperature range of liquid state is in contrast to the highly concentrated electrolytes that suffer from salt precipitation at low temperatures[35]; simultaneously, the cost issue of the salt-concentrated method can be alleviated[29].

**Solution structure analysis of ZES.** The formation mechanism of the ZES was explored by various spectrum analyses. The Raman bands at 3354 and 3157 cm$^{-1}$ in solid Ace correspond to the asymmetric and symmetric NH stretching, respectively (Fig. 1a, left). Upon the introduction of Zn(TFSI)$_2$, the 3354 cm$^{-1}$ band moves to 3380 cm$^{-1}$ while the 3157 cm$^{-1}$ band disappears, suggesting the breaking of H-bonding between Ace molecules. The SO$_3$ and CF$_3$ groups of TFSI$^-$ are proved to be sensitive to the cation–anion and anion–solvent interactions[36,37]. Once eutectic liquid formed, there is a strong interaction between the NH$_2$ group on Ace and the SO$_2$ group on Zn(TFSI)$_2$, as implied by the overlap of the bands at 1150 cm$^{-1}$ and 1128 cm$^{-1}$ (Fig. 1a, right)[38,39]. From fourier transform infrared spectroscopy (FTIR) spectra of ZESs (the left panel of Fig. 1b), an evident change appears at the C=O stretching region of Ace, where the 1665 cm$^{-1}$ band red-shifts slightly to 1654 cm$^{-1}$ accompanied by obvious broadening in comparison to the pristine Ace. This is in line with the formation of metal-oxygen coordination between Zn$^{2+}$ and C=O group (Fig. 1e)[39–41]. These intermolecular interactions between components jointly weaken the respective bonds of pristine components, resulting in eutectic solutions (for more details, see Supplementary Figs. 4, 5).

For the highly concentrated electrolytes proposed for LIBs, organic anions tend to coordinate to Li$^+$ and exist dominantly in associated states (i.e., contact ion pairs or ionic aggregates), which effectively tunes the molecular frontier orbit properties of the electrolyte solutions[42]. In theory, this strategy can also be anticipated in multivalent metal electrochemistries, but no one has yet achieved it partially due to shortage of reliable electrolyte systems at present. Inspirationally, the associated Zn$^{2+}$-TFSI$^-$ states have been found by reasonably introducing a neutral ligand (i.e., Ace) to create anion-containing Zn$^{2+}$ species in this work, which is verified in detail below Fig. 1b (right panel) compares the shift of the $v_s$ (SNS) peak (TFSI$^-$)[43,44] as the salt concentration increases, with crystalline Zn(TFSI)$_2$ as the reference (bottom trace). Apparently, this vibration mode is rather susceptible to the change of the TFSI$^-$ environment[42], slightly drifting from 742.7 cm$^{-1}$ at 1:9 to 741.9 cm$^{-1}$ at 1:4. Essentially, the latter is identical to that in crystal lattice (741.6 cm$^{-1}$), indicative of a possible pronounced interionic attraction in ZESs[43]. Turning to the Raman vibration mode of TFSI$^-$ at the same region (Fig. 1c), a deconvolution analysis shows that the Raman band consists of three modes at 740, 744, and 748/747 cm$^{-1}$, arising from free anions (FA)-(#Zn$^{2+}$ = 0), loose ion pairs (LIP)-(#Zn$^{2+}$ = 1), and intimate ion pairs (IIP)-(#Zn$^{2+}$ = 1), respectively[22,37,45]. In all cases of the ZES system, albeit without obvious ionic aggregates (AGG; the anions are coordinated to two or more cations)[37], the ubiquitous presence of cation–anion coordination can be identified. In 1:7 and 1:9 solutions, the majority of TFSI$^-$ anions exist as long-lived LIPs, suggesting the dominant monomeric Zn species coordinated by TFSI$^-$, while the ionic association becomes stronger with more IIPs formed at relatively higher salt

contents (1:4 and 1:5 solutions) (Fig. 1d). Effects related to salt concentration are also imposed on drastic variation in viscosity and ion conductivity (Supplementary Fig. 13c).

The high-resolution mass spectra (HRMS) of ZESs testify the existence of LIPs and IIPs in all given ratios. Typically, distinct signals of various cationic TFSI$^-$-containing complexes ([ZnTFSI(Ace)]$^+$ at $m/z = 403$, [ZnTFSI(Ace)$_2$]$^+$ at $m/z = 462$, and [ZnTFSI(Ace)$_3$]$^+$ at $m/z = 521$) can be detected, but without evidence of free Zn$^{2+}$ ions (Supplementary Fig. 6). Moreover, the variation trend of these cationic peak intensities qualitatively indicates a more pronounced ionic association upon increasing the Zn-salt content, in line with the above Raman results (Fig. 1c, d). It should be noted that the only anionic species of TFSI$^-$ found in HRMS suggests a low possibility of anionic Zn complexes (monomeric) with more than two associated TFSI$^-$ anions (Supplementary Fig. 7).

Theoretical simulations were performed to further identify the ion speciation of ZESs. In both cases of mixtures (1:7 and 1:4), molecular dynamics (MD) simulations predict a competition between the Ace and TFSI$^-$ for coordination to Zn$^{2+}$ cations (Supplementary Fig. 8). For the 1:7 ratio, one TFSI$^-$ anion (on average) could be observed in each Zn$^{2+}$ primary solvation sheath, typically in the form of the [ZnTFSI(Ace)$_2$]$^+$ solvate (Supplementary Fig. 8a, b). However, in 1:4 ZES, where only four Ace molecules per Zn(TFSI)$_2$ are involved in eutectic solution formation, a lower Ace population is available for Zn$^{2+}$ solvation and H-bonding with TFSI$^-$ anions simultaneously[42]; instead, more TFSI$^-$ anions enter the Zn$^{2+}$ solvation sheath (Supplementary Fig. 8c, d). The fraction of neutral Zn complexes coordinated by two TFSI$^-$ anions is thus expected to increase, whereas no three TFSI$^-$ coordination case was found (Supplementary Fig. 8d). Apparently, ZES is a system featured with the existence of anion-associated Zn solvates, and the ionic interplay strength can be tuned through simple regulation of the Zn(TFSI)$_2$/Ace ratio. By virtue of structural flexibility, the TFSI$^-$ anion may be coordinated in varying ways to Zn$^{2+}$ cations[37], incurring dynamic equilibria of cationic or neutral species with various configurations (Fig. 1g).

Given the fact that TFSI$^-$ is more likely to form bidentate coordination to a single cation than other common anions (i.e., PF$_4^-$, ClO$_4^-$, and BF$_4^-$)[46], the local atomic configurations of Zn complexes were investigated theoretically. The density functional theory (DFT) geometry optimization of [ZnTFSI(Ace)$_n$]$^+$ complexes verifies the preference of the C=O group of Ace and both two O atoms of TFSI$^-$ for the coordination with the central Zn$^{2+}$ cation (Supplementary Figs. 9, 11). The [ZnTFSI(Ace)$_2$]$^+$ structure with bidentate coordination by TFSI$^-$ possesses the most uniform molecular electrostatic potential energy surface distribution along with relatively low total binding energy (Fig. 1f and Supplementary Figs. 9, 11, 12), in reasonable agreement with the predominant signal of cationic species observed from HRMS. Note that the steric-hindrance effect caused by the bulky TFSI$^-$ also dictates the identity of solution species. This can be reflected by the absence of anionic Zn solvates and lower tendency of bidentate coordination in [ZnTFSI$_2$(Ace)$_n$] complexes (Supplementary Fig. 10).

**Electrochemical and ion-transport properties of the ZES.** On the optimization of electrolytes, the ZES with a molar ratio of 1:7 was found to possess a relatively high ionic conductivity (0.31 mS cm$^{-1}$), a low viscosity (0.789 Pa·s) at 25 °C, as well as an optimum Zn/Zn$^{2+}$ redox activity (Supplementary Fig. 13 and Supplementary Table 3). Taking the physical/chemical properties and cost factors into consideration, we chose the molar ratio of 1:7 as the main research object (for the selection of the control

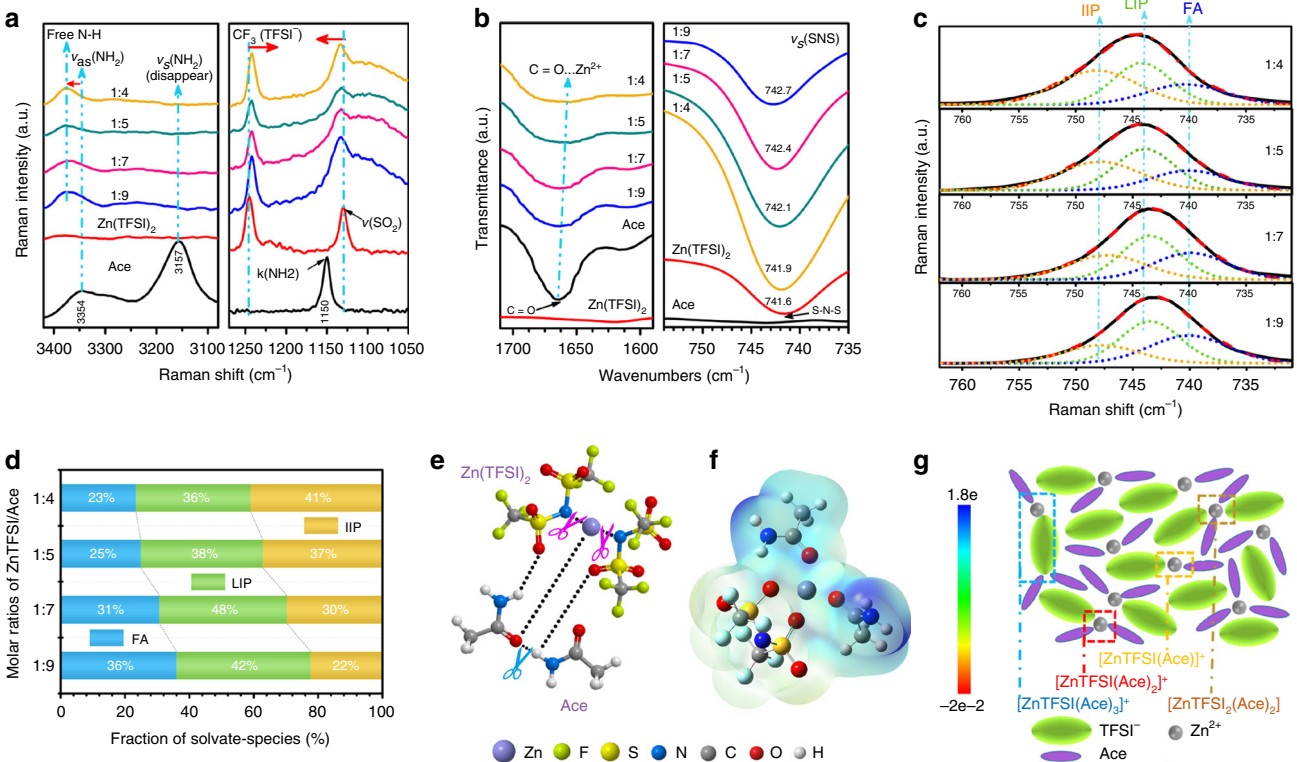

**Fig. 1** Structure analysis of ZESs and identity of the ionic species. **a** Raman, **b** FTIR, and **c** Fitted Raman spectra of ZESs with different $Zn(TFSI)_2$/Ace molar ratios (1:9–1:4). Solid and dashed lines denote experimental spectra and fitting curves, respectively. **d** Solvate species distribution in ZESs (free anions (FA), loose ion pairs (LIP) and intimate ion pairs (IIP)), all obtained from the fitted Raman spectra. **e** Schematic diagram of the interplay among $Zn^{2+}$, $TFSI^−$, and Ace to form eutectic solutions. **f** Molecular electrostatic potential energy surface of $[ZnTFSI(Ace)_2]^+$ ($C_2$-O-Π, bidentate coordination of $TFSI^−$) based on density functional theory (DFT) simulation. Electron density from total self-consistent-field (SCF) density (isoval = 0.001). **g** Illustration of representative environment of active Zn species within the ZES.

group see Supplementary Figs. 14, 15). Supplementary Fig. 16 displays the voltametric response of ZES as compared with an aqueous electrolyte of 1 M $Zn(TFSI)_2$. It is evident that due to the water-splitting reaction, the potential window of 1 M $Zn(TFSI)_2$ is restricted to 1.9 V (vs. $Zn/Zn^{2+}$). In contrast, the ZES provides an expanded anodic stability limit of 2.4 V (vs. $Zn/Zn^{2+}$), also outperforming those of DESs formed by other common Zn salts (e.g., $Zn(ClO_4)_2$, $Zn(CH_3COO)_2$, and $Zn(BF_4)_2$) (Supplementary Fig. 17). The features regarding thermal and electrochemical stabilities allow ZES to be coupled with a wide range of high-voltage cathodes and to work at elevated temperatures, enabling elaborate optimizations of operating conditions in batteries.

In addition, the ZES exhibits a much higher $Zn^{2+}$ transference number (0.572, Supplementary Fig. 18a) as compared with those of other available Zn liquid electrolytes (0.2–0.4)[47]. This effective migration of metal cations is most likely accounted for by the peculiar cationic Zn solvates with tethered anions[48], and the resulting limited transport for negative charge carriers, which is analogous to the observations in highly concentrated electrolytes[29,49]. Furthermore, the high $Zn^{2+}$ transference number also implies that the ion-transport manner in ZESs differs from those observed in the conventional dilute electrolytes; the active $Zn^{2+}$ species might obey underlying hopping-type ion-transport mechanisms ($Zn^{2+}$ ions move from one anion to another through Lewis basic sites on $TFSI^−$ with the aid of Ace matrix)[29,49]. As shown in Supplementary Fig. 18b, the diffusion coefficient of the active Zn species through ZES is $1.66 \times 10^{-6}$ $cm^2 s^{-1}$, exceeding ionic liquid-based electrolytes reported[50,51]. These remarkable kinetic properties make for powering high-rate devices[50,52,53].

**High Zn/$Zn^{2+}$ reversibility and uniform Zn deposits.** The CE of Zn plating/stripping, the most critical parameter responsible for the redox reversibility, was first investigated in Zn/Ti cells with a galvanostatic capacity of 1 mAh cm$^{-2}$ (0.5 mA cm$^{-2}$) (Fig. 2a; Supplementary Fig. 19). Of note, the CE of the first 10 cycles in ZES rises gradually to above 98.0%; instead, the inferior CE of <70% was obtained in 1 M $Zn(TFSI)_2$ (Supplementary Fig. 20a, b) under identical conditions, which could be ascribed to the severe parasitic reactions that simultaneously occurred during Zn deposition[17], along with uncontrolled dendrites (Supplementary Fig. 20d)[12,54]. Interestingly, an overpotential of 0.185 V is required for the 1st cycle in ZES while roughly 0.1 V needed in the following cycles (Fig. 2a, green circle), which suggests the increase in surface area as well as the progressively improved stability induced by stepwise generation of the in situ formed interphase[55,56]. The relatively lower CE of ZES in initial cycles might originate from the consumption of active $Zn^{2+}$ for such interfacial activation. Additional support for our hypothesis comes from the post-mortem scanning electron microscopy (SEM) observation. It is apparent that a protective coating layer formed on the Ti surface deposited by flat and dense Zn (Supplementary Fig. 20c), essentially differing from the tanglesome deposits in 1 M $Zn(TFSI)_2$ (Supplementary Fig. 20d). As another reliable method[16,57], cyclic voltammetry (CV) was further applied to evaluate CE of ZES at an average deposition capacity ~0.61 mAh cm$^{-2}$ (Fig. 2b). Corresponding chronocoulometry curves (Fig. 2c) reveal that the plating/stripping is highly reversible; the CE approaches 100% after the initial 30 conditioning cycles (an average CE of 99.7% for 200 cycles; see Supplementary Fig. 21). From this aspect, compared with other reported Zn electrolytes

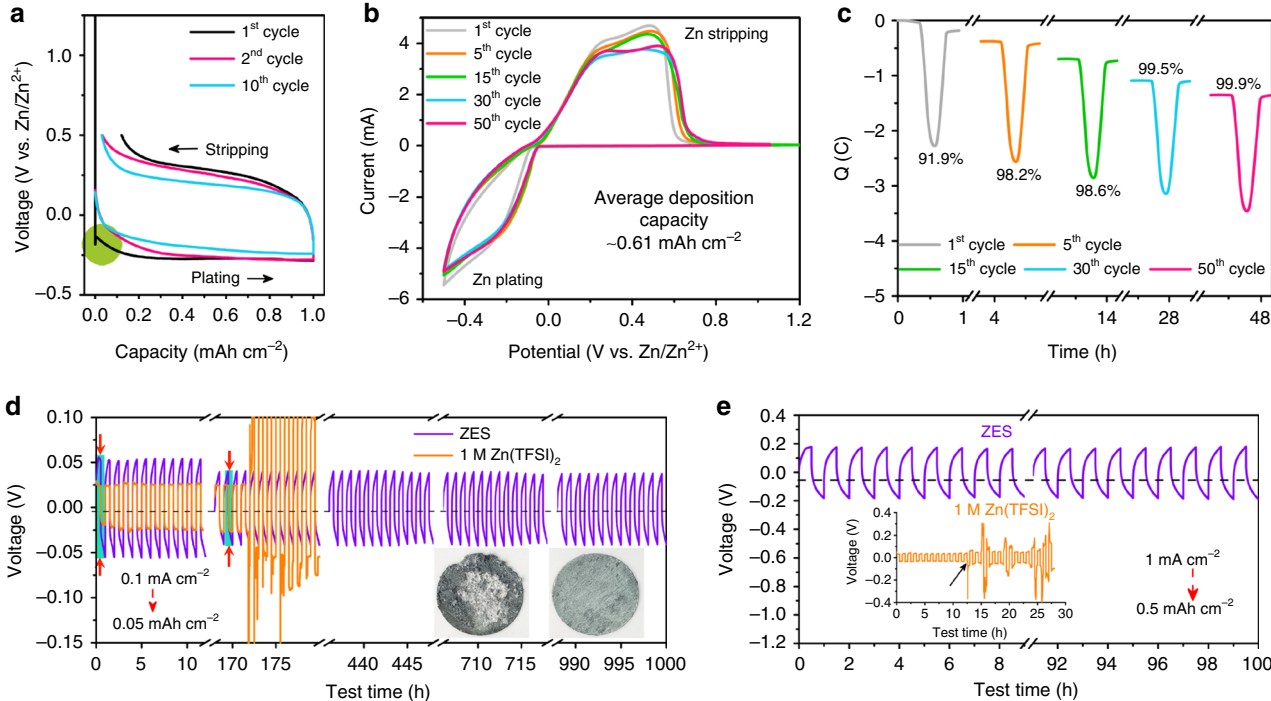

**Fig. 2** Zn plating/stripping behaviors in ZES. **a** Voltage profiles of galvanostatic Zn plating/stripping with the maximum oxidation potential of 0.5 V (vs. Zn/Zn²⁺) in ZES at a rate of 0.5 mA cm⁻² (1.0 mAh cm⁻²). The working and counter electrodes are Ti and Zn, respectively. **b** Cyclic voltammetry (CV) curves of Zn plating/stripping in ZES at a scan rate of 1 mV s⁻¹ with a potential range of −0.5–1.2 V and an average deposition capacity of ~0.61 mAh cm⁻². The working and counter electrodes are Ti and Zn, respectively. **c** Chronocoulometry curves of Zn plating/stripping in ZES based on CV. Voltage responses of Zn/Zn symmetric cells **d** in ZES and 1 M Zn(TFSI)₂ electrolytes at 0.1 mA cm⁻² (0.05 mAh cm⁻² for each half cycle) for 1000 cycles (insets: the optical images of the cycled Zn after 180 cycles in 1 M Zn(TFSI)₂ (left) and 2000 cycles in ZES (right)), and **e** in ZES electrolyte (inset: in 1 M Zn(TFSI)₂ electrolyte) at 1 mA cm⁻² (0.5 mAh cm⁻² for each half cycle).

(see Supplementary Table 4)[16,17,58], ZES provides a more promising route for the realization of secondary Zn-metal cells to charge for hundreds of times, especially when the excess of Zn anode is limited.

The superior performance of ZES for supporting the Zn anode was also demonstrated under galvanostatic conditions in a Zn/Zn symmetric configuration. Despite of a slightly larger polarization, all cells using ZES exhibit more sustainable electrochemical cycling in contrast to those with 1 M Zn(TFSI)₂. As viewed from Fig. 2d, the overpotential experienced a gradual decrease (from 55 to 39 mV) upon cycling at 0.1 mA cm⁻² in ZES, conforming to the formation process of SEI. As the rate was raised to 0.5 mA cm⁻², the same cell with ZES continued to operate steadily for another 1000 h (Supplementary Fig. 22a). Note that the surface morphology of cycled Zn in ZES is visually uniform (Fig. 2d inset right and Supplementary Fig. 22c), while characteristic Zn protrusions are shown in the case using 1 M Zn(TFSI)₂ (Fig. 2d inset left and Supplementary Fig. 22b). Even cycled at an elevated rate of 1 mA cm⁻² and a capacity of 0.5 mAh cm⁻², the Zn/Zn cell with ZES also maintains an impressive stability without voltage fluctuation, which lays the foundation for designing high-rate ZIBs. In sharp contrast, an erratic voltage response with the rapidly rising overpotential occurred after only 15 cycles in 1 M Zn(TFSI)₂ (Fig. 2e inset). Given that side-reactions at the electrolyte–electrode interface are considered to be more competitive at relatively low rates[12], further interrogation of the ZES electrolyte was carried out at 0.01−0.05 mA cm⁻² (charge/discharge interval being extended to 10 h, Supplementary Fig. 23). The Zn/Zn²⁺ redox reactions remain reversible and steady with a cycling life over 200 h.

The Zn-electrolyte interface stability has been detected by the sensitive electrochemical impedance spectroscopy (EIS). The charge-transfer resistance of symmetric Zn/Zn cell using 1 M Zn(TFSI)₂ keeps increasing with cycling, reaching 356 ohm after 15 cycle (Supplementary Fig. 24). In the case of ZES, a much better interfacial compatibility can be obtained and the charge-transfer resistance maintains steady after 15th cycles. This significant difference is ascribed to, in part, the competing H₂ evolution reaction (Supplementary Fig. 25a) and the accumulation of undesirable passivating byproducts (such as Zn(OH)₂, xZnCO₃•yZn(OH)₂•zH₂O and ZnO) on Zn anode surface upon cycling in 1 M Zn(TFSI)₂ (Supplementary Fig. 25b). By contrast, the cycled Zn obtained from ZES presents well-defined X-ray diffraction (XRD) peaks, agreeing well with the Zn reference (PDF#99-0110). The time evolution of interface resistance under open circuit conditions further verifies the chemical stability of the metallic Zn in ZES (Supplementary Fig. 26).

The utilization of ZESs along with the possible in situ SEI layer can have a large impact on the Zn deposition. Upon deposition capacity of 0.5 mAh cm⁻², loose structures with uncontrolled dendritic Zn growth appeared in 1 M Zn(TFSI)₂ (Fig. 3a, c), which surely accounts for the low CE (Supplementary Fig. 20a). This is not the case for our ZES electrolyte, as SEM images in Fig. 3d, e clearly show dendrite-free and smooth Zn deposits, even at higher capacity of 2.5 mAh cm⁻². Based on the cross-sectional views (Fig. 3f, g), thickness of the deposited Zn layer is about 5.2 μm, in line with the expected value 4.3 μm for capacity of 2.5 mAh cm⁻², which represents the dense Zn coating by electrodeposition in ZES. Notably, with increasing current densities (Fig. 3e, f), the Zn deposits become more compact (theoretical/actual thickness: 1.7/2.3 at 0.5 mA cm⁻² less than 4.3/5.2 at 0.25 mA cm⁻²) while the particle size decreases slightly, following the classical nucleation theory[56]. It is also visible that the Zn anode after deposition is covered by a thin surface layer

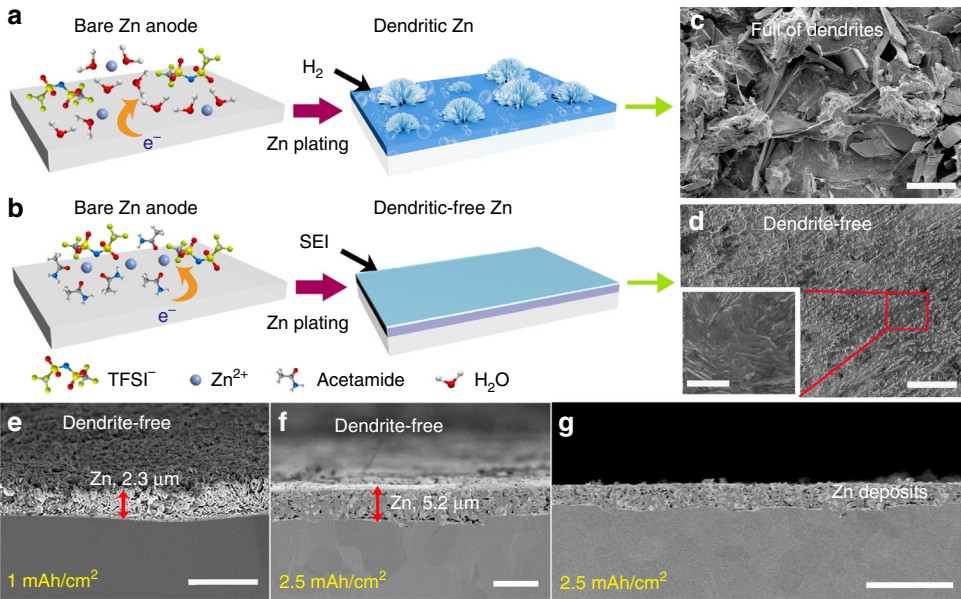

**Fig. 3** Effect of ZES and as-obtained SEI layer on Zn deposition. **a** Zn dendrite growth along with $H_2$ evolution observed in 1 M Zn(TFSI)$_2$ and **b** SEI-regulated uniform Zn deposition in ZES. SEM images of Zn deposits using **c** 1 M Zn(TFSI)$_2$ and **d** ZES electrolytes at 1 mA cm$^{-2}$ (0.5 mAh cm$^{-2}$). **e–g** Cross-sectional SEM images of Zn deposits which were obtained in ZES electrolyte with 1 mAh cm$^{-2}$ (0.25 mA cm$^{-2}$) (**e**) and 2.5 mAh cm$^{-2}$ (0.5 mA cm$^{-2}$) (**f**) Zn on Zn substrate, respectively. **g** A lower-magnification image of panel (**f**) showing a large area of uniform deposition. Scale bar: 50 μm for (**c**), (**d**); 2 μm for (**e**); 5 μm for (**f**); 20 μm for (**g**) and inset in (**d**).

(Fig. 3d inset), also corresponding to the surface modification. Thus, it is reasonable to assume that this additional Zn-electrolyte interphase dictates the reversible Zn/Zn$^{2+}$ redox with efficient Zn$^{2+}$ transport and deposition (Fig. 3b).

**Formation mechanism and chemical composition analysis of the SEI layer on Zn anode.** A uniform SEI can be rationally constructed by introducing selected elements and/or compounds (such as F-donating salts and solvents) that decompose beneficially on alkaline metal anodes[59], but due to the higher redox potential of the Zn/Zn$^{2+}$ couple (−0.76 V vs. NHE) than that of free TFSI$^{-}$ (−0.87 V vs. Zn/Zn$^{2+}$)[42], the reductive TFSI$^{-}$ decomposition can hardly take place before Zn deposition. In our case, by virtue of the intrinsic ion-association network in present eutectic liquid, a marked change of the TFSI$^{-}$ coordination environment has been confirmed (Fig. 1), making the anion-derived SEI formation for metallic Zn possible. DFT calculations demonstrate the altered reduction potential of TFSI$^{-}$ by its intimate interaction with Zn$^{2+}$ (Fig. 4a). The Zn$^{2+}$-TFSI$^{-}$ complexes become reductively unstable below 0.37 V (vs. Zn/Zn$^{2+}$), which is substantially higher than the reduction potential for the isolated TFSI$^{-}$, corroborating the preferential decomposition of TFSI$^{-}$ over Zn$^{2+}$ reduction.

In support of the above mechanism, X-ray photoelectron spectroscopy (XPS) and Raman analyses were implemented to experimentally probe the existence of the in situ formed interphase. In the F 1s spectra from XPS (Fig. 4b, the C 1s spectrum can be seen in Supplementary Figs. 27a, 28), except for the C–F component arising from the residual Zn(TFSI)$_2$ salt, we also observed the presence of ZnF$_2$ (684.5 eV) on the cycled Zn[60], in perfect accordance with the DFT calculation. For the S 2p spectrum shown in Fig. 4c and Supplementary Fig. 27d, a new peak associated with sulfide appears at 161.9 eV[45], further verifying the decomposition of TFSI$^{-}$. Encouragingly, from the Raman analysis, a strong characteristic peak assigned to ZnF$_2$ at 522 cm$^{-1}$ can be clearly detected[61] (Fig. 4d). On the other hand, the Zn anode cycled in 1 M Zn(TFSI)$_2$ displays two obvious

Raman peaks at 437 and 563 cm$^{-1}$, related to the formation of ZnO[62].

FTIR investigations provide additional insights into the chemical features of the Zn-compatible SEI. Compare with pure Zn, the surface of the SEI-coated Zn is enriched with organic functional groups (Supplementary Fig. 30). A blue shift in the C–F, S=O and C–N functional groups can be found for the SEI layer (Supplementary Table 5) compared with the ZES electrolyte. Overall, the agreement between spectral characterization and theoretical calculation suggests that the unique anion-containing Zn species in ZES enable TFSI$^{-}$ to reductively decompose and participate in the SEI formation with major components involving ZnF$_2$, S/N-rich organic compounds and/or their derivatives. Unlike the passivating layers on the surface of multivalent metals (Zn, Mg, and Ca, etc.)[23,24], Zn$^{2+}$ is able to penetrate such anion-derived SEI layer, whose ionic conductivity is calculated to be $2.36 \times 10^{-6}$ S cm$^{-1}$ (Supplementary Fig. 31), higher than that of the artificial interphase fabricated on Mg anode ($1.19 \times 10^{-6}$ S cm$^{-1}$)[23]. An additional evidence for the effective Zn$^{2+}$ diffusion through SEI can be provided by the low activation energy of 49.7 kJ mol$^{-1}$ obtained by temperature-dependent impedance measurements (Fig. 4e and Supplementary Fig. 32)[25,63], which is comparable to that of Li$^{+}$ diffusion across the typical SEI formed in LiPF$_6$/EC/EMC (51 kJ mol$^{-1}$)[25]. Furthermore, several metal fluorides ($M_xF_y$, $M$ = Li, Zn, Cu, and Al) have been well acknowledged as main rigid-frame materials for protecting metal anodes, since they can guide the metal nucleation and effectively inhibit the growth of dendrites[64–66]. Meanwhile, the S/N-rich organic compounds could provide sufficient ion channels for Zn$^{2+}$ transport, and their flexibility will accommodate volume changes caused by Zn plating/stripping[67,68].

In view of the element distribution of the SEI, two additional surface-sensitive techniques, time-of-flight secondary-ion mass spectrometry (TOF-SIMS) and XPS spectra at various sputtering depths were used. As shown in XPS spectra at various sputtering depths (0, 7, and 15 nm) on the SEI accumulated on Zn (Fig. 4f),

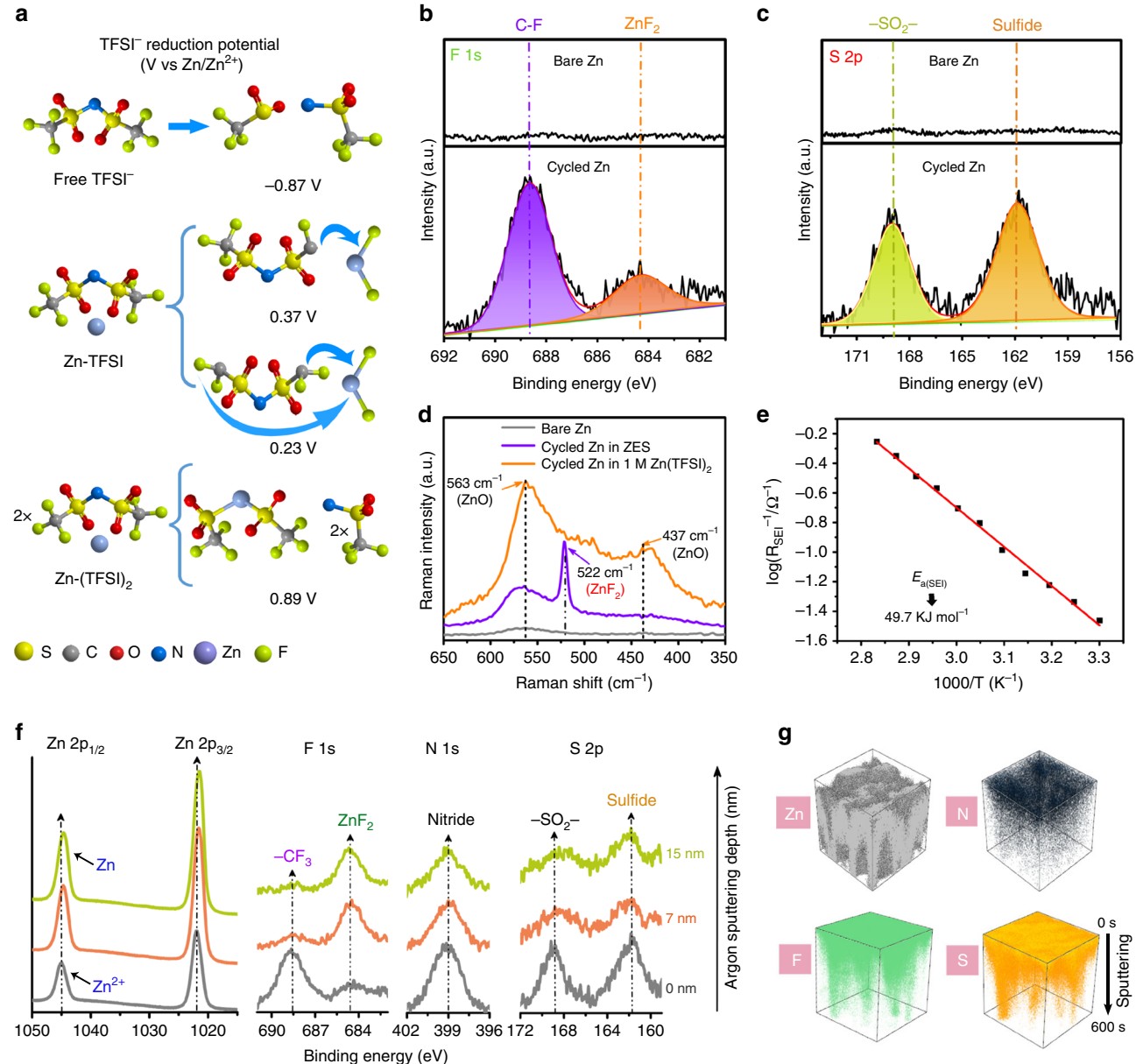

**Fig. 4** Experimental and theoretical investigations on the existence and the composition of the Zn-compatible SEI layer. **a** Predicted reduction potentials by DFT calculations. XPS spectral regions for **b** F 1s and **c** S 2p of the surface of bare Zn (top) and after cycled Zn (bottom), respectively. **d** Raman spectra of the cycled Zn anode in ZES and 1 M Zn(TFSI)$_2$. **e** Arrhenius behavior of the reciprocal resistances corresponding to interfacial components and the activation energy derived for the in situ formed SEI. **f** XPS spectral regions for Zn 2p, F 1s, N 1s, and S 2p at various argon (Ar$^+$) sputtering depths on the SEI accumulated on Zn substrate. **g** Three-dimensional view of Zn, N, F, and S elements distributions of SEI in the time-of-flight secondary-ion mass spectrometry (TOF-SIMS) sputtered volumes. The SEI-coated Zn anode was obtained after 20 cycles of galvanostatic plating/stripping in ZES electrolyte (Zn/Zn cells at 0.5 mA cm$^{-2}$ with a capacity of 1 mAh cm$^{-2}$ for each half cycle).

the signal of metallic Zn at 1044.63 eV can be observed with the depth increased to ca. 15 nm (Fig. 4g and Supplementary Fig. 29), which defines the thickness of SEI layer. Moreover, as the etching depth increased, the intensity of ZnF$_2$ increases gradually while those of sulfides and nitrides decrease, indicating that ZnF$_2$ mainly exists in the inner SEI region and S/N-rich organic compounds are mainly distributed in the outer SEI layer. Besides, TOF-SIMS results (Fig. 4f and Supplementary Fig. 33) combined with the energy-dispersive spectroscopy (EDS) analyses (Supplementary Fig. 34) further exhibit an even distribution of Zn, F, N, and S elements on the cycled Zn surface, implying the uniformity of the SEI layer.

**The validity of the SEI layer in various electrolyte systems.** More direct evidence for modulated Zn plating/stripping enabled by this SEI was obtained from in situ optical visualization observations of Zn deposition (in a home-made cell, Supplementary Fig. 35a). Not surprisingly, in 1 M Zn(TFSI)$_2$, rather uneven Zn electrodeposits and copious air bubbles were observed as early as 5 min after the inception of deposition (Fig. 5a and Supplementary Fig. 35b). For the cell using ZES, uniform and compact Zn deposits can be achieved at capacity of 5 mAh cm$^{-2}$ (10 mA cm$^{-2}$) (Fig. 5b). We further investigated the availability of the SEI layer in the routine aqueous electrolyte. As is shown in Fig. 5c, the gas generation disappears when such SEI-coated Zn

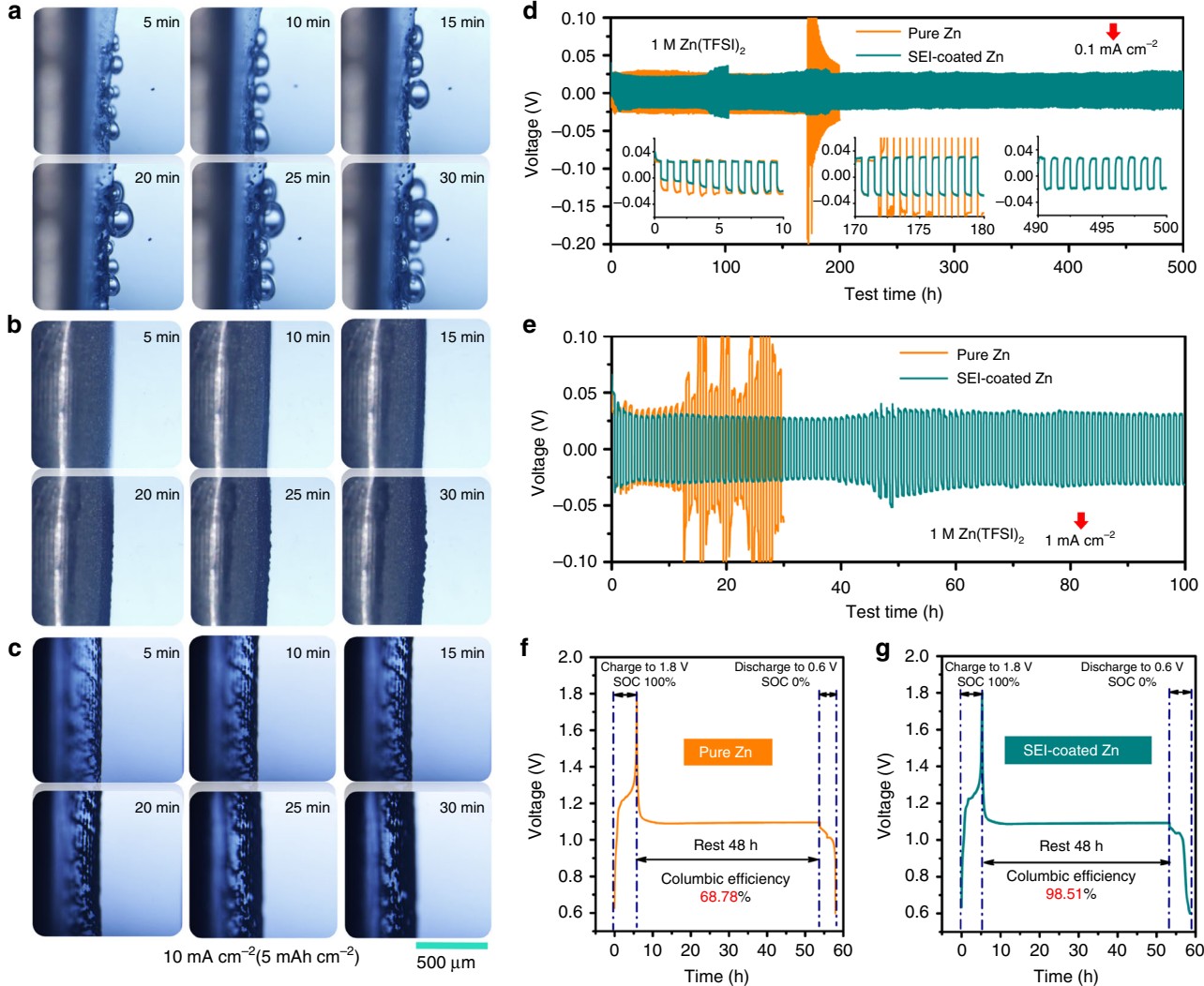

**Fig. 5** The validity of the SEI layer for Zn electrochemistry. **a–c** In situ investigations of the Zn deposition by optical microscopy in Zn/Zn cells. Images of the Zn-electrolyte interface region in **a** 1 M Zn(TFSI)$_2$ and **b** ZES. **c** Images of SEI-coated Zn-electrolyte interface region using 1 M Zn(TFSI)$_2$. The deposition current density is 10 mA cm$^{-2}$ with the areal capacity of 5 mAh cm$^{-2}$. Galvanostatic cycling performance of symmetric Zn/Zn cells tests using pure and SEI-coated Zn coupled with 1 M Zn(TFSI)$_2$ at **d** low rate of 0.1 mA cm$^{-2}$ (0.05 mAh cm$^{-2}$ for each half cycle) and **e** high rate of 1 mA cm$^{-2}$ (0.5 mAh cm$^{-2}$ for each half cycle). The Zn/V$_2$O$_5$ cells in 1 M Zn(TFSI)$_2$ using **f** pure Zn anode and **g** SEI-coated Zn anode were first fully charged to 1.8 V at 20 mA g$^{-1}$ (based on active materials of cathode), respectively, and then the cells were rested at 100% stage of charge (SOC) for 48 h, followed by full discharging.

anodes obtained in ZES were reassembled in 1 M Zn(TFSI)$_2$. Particularly, these surface-modified Zn anode exhibits extended cycling life and lower polarization at both low (0.1 mA cm$^{-2}$) and high (1 mA cm$^{-2}$) rates (Fig. 5d, e). Indeed, Zn surface after deposition remain visibly flat (Supplementary Fig. 35c), which emphasizes a strong correlation between SEI and dendrite-free Zn deposition.

The effect of the SEI on parasitic reactions was evaluated by monitoring the open circuit-voltage decay of fully charged Zn/V$_2$O$_5$ cells with 1 M Zn(TFSI)$_2$ and then discharging after 48 h of storage. 97.8% of the original capacity was retained (Fig. 5f) in cell using SEI-protected Zn anode, exceeding 68.78% using an untreated Zn anode. Obviously, similar to the function of the anode SEI layer obtained in Li-metal anodes, this rigid-flexible coupling SEI formed on Zn surface can eliminate direct contact between the active anode and electrolyte, thereby inhibiting the interface side-reactions (such as H$_2$ evolution and passivation) effectively during storage. More importantly, once the SEI forms, the coated Zn surface is functionalized by stable and favorable

Zn$^{2+}$ transport with low-diffusion barrier, which can facilitate reversible Zn stripping/plating even applicable for implanted aqueous electrolytes. Note that ZES exhibits lower polarization and better stability for Zn/Zn$^{2+}$ reactions compared with DESs based on other Zn salts (Supplementary Fig. 36), demonstrating the TFSI$^-$-induced SEI formation mechanism and the uniqueness of the SEI composition. To the best of our knowledge, the in situ constructed effective SEI has not been reported in ZIBs. This strategy may also be helpful to the development of other MIBs electrolytes that are compatible with their corresponding metal anodes.

**Highly stable ZIBs coupled with cathodes (V$_2$O$_5$ and Mo$_6$S$_8$) in the ZES electrolyte**. Finally, we explored applications of the ZES electrolyte in ZIBs composed of Zn anode and V$_2$O$_5$ cathode (Fig. 6a). CV profiles of the cell after initial cycle activation are almost overlapped. For comparison, a co-intercalation of Zn$^{2+}$ and hydrated protons (H$_3$O$^+$) was observed in the cell that

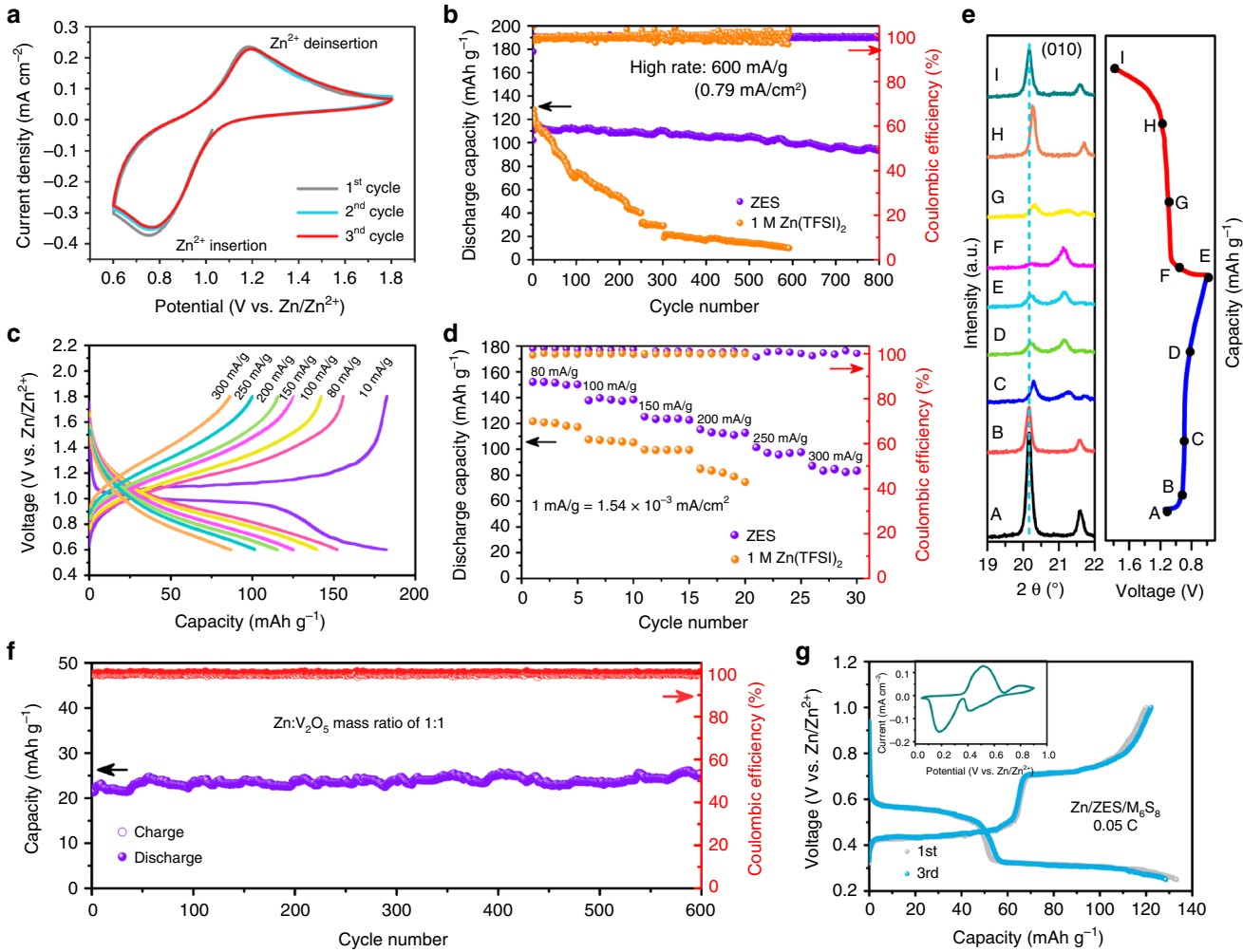

**Fig. 6** Electrochemical properties of ZIBs. **a** Typical CV curves of the Zn/V$_2$O$_5$ cell using ZES at a scan rate of 0.5 mV s$^{-1}$. **b** Charge/discharge cycling performance and CE of the Zn/V$_2$O$_5$ cells with ZES (after activation under 1 A g$^{-1}$) and 1 M Zn(TFSI)$_2$ electrolytes at 600 mA g$^{-1}$ (0.79 mA cm$^{-2}$). **c** Charge/discharge curves at various current densities in ZES. **d** Rate performance of ZES and 1 M Zn(TFSI)$_2$ electrolytes. **e** XRD patterns of the V$_2$O$_5$ cathode at different voltage states of the first cycle in ZES (10 mA g$^{-1}$). **f** Long-term cycling performance of the Zn//ZES//V$_2$O$_5$ cell with the Zn:V$_2$O$_5$ mass ratio of 1:1 at 8.43 mA cm$^{-2}$ (after activation under same rate; the capacity is calculated based on the total mass of cathode and anode). **g** Typical galvanostatic charge/discharge profiles and CV curves (inset) of the Zn/Mo$_6$S$_8$ cell with ZES electrolyte. The current densities are calculated on the activated materials of cathode.

contains 1 M Zn(TFSI)$_2$ as electrolyte (Supplementary Fig. 37)[69]. From the CV measurement of the Zn/V$_2$O$_5$ cell at different scan rates, the diffusion-controlled reaction process can be certified in ZES (Supplementary Fig. 38). As expectedly, the cyclic stability of Zn/V$_2$O$_5$ cells using ZES outperforms their aqueous counterparts at each current density (Fig. 6b and Supplementary Figs. 39, 41). Unlike 1 M Zn(TFSI)$_2$, ZES can be cycled without over-charging even at a relatively low rate (10 mA g$^{-1}$) (Supplementary Fig. 40). At a high rate of 200 mA g$^{-1}$ (~34 min rate), the V$_2$O$_5$ cathode with ZES delivers an excellent stability with a high capacity retention of 91.3% with a high CE ~99.34% for 100 cycles (Supplementary Fig. 41a). Besides, there is only a slight change of the cell overpotential throughout the cycling process (Supplementary Fig. 41b). Considering that the generation of a stabilized SEI extremely relies on the amount of charge passed through the cell[54], we pre-activated the Zn/V$_2$O$_5$ cell at 1 A g$^{-1}$ to accelerate the SEI growth on anode, and then tested its long-cycle stability at the rate of 600 mA g$^{-1}$. Such a cell exhibits a highly reversible specific capacity of nearly 110 mAh g$^{-1}$ (based on the mass of V$_2$O$_5$); 92.8% of initial capacity could be retained over prolonged 800 cycles, along with a high average CE of 99.9%

(Supplementary Fig. 42). In sharp contrast, the capacity of the cell with 1 M Zn(TFSI)$_2$ rapidly decayed to 61.9 mAh g$^{-1}$ (capacity retention < 50%) after only 150 cycles, which is mainly ascribed to the formation of the insulating passivation layer on Zn anode (Supplementary Fig. 25) that blocks the Zn$^{2+}$ interfacial transport, and the resulting increase in polarization[10,23,26]. The potential of ZES for power-type ZIBs is further evidenced by the attractive rate capability with elevating current density from 80 to 300 mA g$^{-1}$ (Fig. 6c, d). This should be contributed by high active Zn$^{2+}$ diffusion coefficient (1.66 × 10$^{-6}$ cm$^2$ s$^{-1}$, Supplementary Fig. 18) and the pseudo-capacitance properties of V$_2$O$_5$ (Fig. 6c).

Although cycling with a low areal capacity has been demonstrated to assist in maintaining a uniform morphology for metallic anodes[70], material loadings must be rationally optimized to yield the truly competitive ZIBs for industrial scenarios[16]. Thus, we have attempted to estimate the utility of the ZES electrolyte on a more practical basis by a full cell with a high-mass-loading V$_2$O$_5$ cathode and a thin Zn foil (20 μm thickness, ~11.7 mAh cm$^{-2}$). When the V$_2$O$_5$ loading is as high as 14.3 mg cm$^{-2}$, the Zn//ZES//V$_2$O$_5$ cell can be cycled still shows stable operation over 600 cycles at a high rate of 8.43 mA cm$^{-2}$ with a

capacity fading of only 0.0035% cycle$^{-1}$ (the capacity retention of 97.89%) (Fig. 6f). In contrast to most of the previously reported ZIBs, wherein much excessive Zn needs to be used for prolonging the cycle life, the mass ratio between Zn and $V_2O_5$ was set to 1:1 in this cell. Based on the total mass of cathode and anode, the capacity is calculated to be 25.5 mAh g$^{-1}$, corresponding to an energy density of 25.8 Wh kg$^{-1}$. In addition, further reducing the Zn:$V_2O_5$ mass ratio to 0.5:1 can provide an improved energy density of 40.9 Wh kg$^{-1}$ (Supplementary Fig. 43). In the case of development of the $Zn^{2+}$-storage cathodes taking into account stability, capacity and operation voltage simultaneously, there is still vast scope for improvements in energy density of ZES-based ZIBs[71]. The ex situ XRD test (Fig. 6e and Supplementary Fig. 44) shows that a highly stable and completely reversible structure evolution occurred on the $V_2O_5$ cathode during charge/discharge processes in ZES. A $Zn/Mo_6S_8$ cell was assembled to demonstrate the versatility of ZES for ZIB applications (Fig. 6g). Such a cell delivers a high discharge capacity of 128.6 mAh g$^{-1}$ (based on the mass of $Mo_6S_8$) after three cycles, which is close to the theoretical value of $Mo_6S_8$ (129 mAh g$^{-1}$)[72]. Two pairs of typical redox peaks correspond well to the two-step $Zn^{2+}$ (de-)intercalation processes, analogous to cases of previously reported[72].

## Discussion

In summary, we have demonstrated that the established in situ SEI protection is a feasible strategy toward rechargeable Zn-metal anodes. Due to the direct coordination between cations and anions in a form of large-size cationic complexes endowed by the ZES, the reductive decomposition of TFSI$^-$ is induced before the Zn deposition during the initial cycling process, allowing a well-defined Zn-compatible SEI layer with a rich content of mechanically rigid $ZnF_2$ and $Zn^{2+}$-permeable organic components. With this interface modulation, dendrite-free and intrinsically stable Zn plating/stripping can be realized at the areal capacity of >2.5 mAh cm$^{-2}$ or even under a common dilute aqueous electrolyte system. Zn//ZES//$V_2O_5$ cells present remarkable electrochemical reversibility (an average CE of ~99.9%, superior to most aqueous ZIBs[9,73,74]) and laudable capacity retention even under rigorous but practically desirable cathode-anode loading conditions. Given the extendibility of this strategy, we envision that this study will provide an unprecedented avenue for tackling the dilemmas raised by the intrinsic properties of multivalent metal anodes, which may lead to the potential fabrication of energy-storage devices.

## Methods

**Preparation of electrolytes and cathodes**. The ZES samples were formed by readily mixing the two components ($Zn(TFSI)_2$ and Ace) with the required molar ratios (the $Zn(TFSI)_2$/Ace molar ratio between 1:4 and 1:9) at room temperature (Supplementary Fig. 1). Homogenous and transparent liquids can be obtained directly after heating the mixtures at 80 °C with gentle stirring 2 h. Subsequently, the electrolytes were stored in a dry atmosphere for further use. The micro-sized $V_2O_5$ material was purchased from Aldrich. The $V_2O_5$ electrodes used here comprise 70 wt% $V_2O_5$, 20 wt% Super P carbon, and 10 wt% polyvinylidene fluoride (PVDF; Sigma). $Mo_6S_8$ (Chevrel phase) was synthesized according to the previously reported method[72]. The $Mo_6S_8$ electrodes were prepared by the same procedure, but $Mo_6S_8$ (80 wt%), Super P carbon (10 wt%), and PVDF (10 wt%), which were mixed and dispersed in N-methyl-2-pyrrolidone and cast on to the Ti current collector (10 μm in thickness). $V_2O_5$ and $Mo_6S_8$ cathodes were punched in the diameter of 1.2 cm (1.1304 cm$^2$) for the full cell tests. The active mass loading for the $V_2O_5$ cathode materials is 1.6 ± 1 mg cm$^{-2}$ for normal tests, while that for the $Mo_6S_8$ cathode is ~1.5 mg cm$^{-2}$. The high active mass loading for the $V_2O_5$ cathode materials is 14.3 mg cm$^{-2}$ (4.1 mAh cm$^{-2}$, 290 mAh g$^{-1}$ for the $V_2O_5$) and the thicknesses of Zn foils are 20 and 10 μm (14.28 and 7.14 mg cm$^{-2}$, respectively) for the practical utility evaluation test.

**Electrochemical measurements**. EIS was performed by an electrochemical working station (VMP-300) over the frequency range 0.1–7 × 10$^6$ Hz with a perturbation amplitude of 5 mV to better investigate the interfacial stability between

Zn metal and different electrolytes. Electrochemical cycling tests in Zn/Zn symmetric cells, Ti/Zn cells, Zn/$V_2O_5$ and Zn/$Mo_6S_8$ cells were conducted in CR2032-type coin cells with LAND testing systems. All cells were assembled in an open environment and a glass fiber with a diameter of 16.5 mm was used as the separator.

**Characterization**. SEM (Hitachi S-4800) was employed to detect the morphologies of Zn deposits on the Zn-metal anodes or the Ti foils. FTIR measurements were carried out on a Perkin-Elmer spectrometer in the transmittance mode. XRD patterns were recorded in a Bruker-AXS Micro-diffractometer (D8 ADVANCE) with Cu-K$_{α1}$ radiation ($\lambda = 1.5405$ Å). Raman spectra were recorded at room temperature using a Thermo Scientific DXRXI system with excitation from an Ar laser at 532 nm. A differential scanning calorimeter (TA, dsc250) was used to evaluate the thermal properties of the electrolytes. Samples are scanned from −80–100 °C at a rate of 5 °C–min$^{-1}$ under a nitrogen atmosphere. An in situ optical microscope from the Olympus Corporation was used to observe the depositional morphology of Zn with different electrolytes in real time in order to study the interfacial stability. XPS was performed on a Thermo Scientific ESCA Lab 250Xi to characterize the surface components. TOF-SIMS (Germany, TOF-SIMS5) was employed to measure the components as a function of depth.

**Calculation methods**. All quantum chemical calculations were performed by applying the DFT method with the B3LYP level and 6–31+G (d, p) basis set using Gaussian 09 program package. The structural optimization was determined by minimizing the energy without imposing molecular symmetry constraints. The binding energy of the anion-containing Zn species were defined as the interaction between different molecule fragments, composed of the interaction between $Zn^{2+}$, TFSI$^-$ and Ace. The binding energy $E$ was calculated according to Eq. (1), the expression as follows:

$$E = E_{total} - nE(X) \tag{1}$$

where $E_{total}$ is the structure total energy, $E(X)$ is the energy of different molecule fragments ($X = Zn^{2+}$, TFSI$^-$, Ace), and $n$ is the number of corresponding molecule fragments according to the different structure configurations

The reduction potentials for the TFSI$^-$ anion with different paths in the solution containing $Zn^{2+}$ were calculated according to Eq. (2), in which the values of the reduction potentials were converted to the $Zn/Zn^{2+}$ scale by subtraction of 3.66 V as discussed extensively elsewhere[42,75]

$$E^0 = -\frac{\Delta G^0_{298K}}{nF} - 3.66 \text{ V} \tag{2}$$

where $\Delta G^0_{298K}$ is the Gibbs energy of the reduction reaction of different paths, $n$ is the transferred electron number, $F$ is Faraday constant.

## Data availability

The datasets generated during the current study are included in this published article (and its supplementary information files) are available from the corresponding author on reasonable request.

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

## Acknowledgements

The authors acknowledge the financial support from the National key R&D Program of China (Grant No. 2018YEB0104300), the Strategic Priority Research Program of the Chinese Academy of Sciences (XDA22010600), the National Natural Science Foundation for Distinguished Young Scholars of China (Grant No. 51625204), National Natural Science Foundation of China (Grant No. 21601195, U1706229), the Key research and development plan of Shangdong Province P. R. China (2018GGX104016), and the Youth Innovation Promotion Association of CAS (2019214, 2016193).

## Author contributions

H.Q. and X.D. contributed equally to the paper. G.C. and J.Z. proposed the concepts. H.Q. designed and carried out the experiments. X.D. and D.Y. performed the theoretical simulations. H.Q., Z.C., Z.H. and J.Z. analyzed the data. Y.W. and J.J. conducted the cross-sectional SEM studies. J.Z. supervised the research. H.Q. wrote the paper with the help from J.Z., G.C. and X.Z. All authors discussed the results and commented on paper.

## Competing interests

The authors declare no competing interests.
