## [Peer Review File · Nature Communications]

Reviewers' comments:

Reviewer #1 (Remarks to the Author):

This manuscript presents highly reversible Zn-deposition/stripping cycling by in-situ formed solid electrolyte interphase (SEI) originated from decomposition of complex of $[\text{Zn-TFSA}(\text{acetamide})_n]^+$ whose coordination environment was analyzed using Raman spectroscopy, FT-IR, and mass spectrometry. DES electrolyte consisted of $\text{Zn}(\text{TFSA})_2$ and acetamide with molar ratio of 1:7 dramatically improved cyclability of Zn electrode without dendrite morphology, whereas routine electrolyte of 1 M $\text{Zn}(\text{TFSA})_2$ showed poor cycling performance with tangle deposition morphology. The obtained high performance comes from ZnF_2 -rich SEI with mechanical rigidity and Zn^{2+} -permeability. The decomposition potential of the SEI was controlled by solvation state so that the SEI was formed at higher potential than that of Zn deposition. The authors achieved a reversible capacity of 51 mA h g⁻¹ in V2O5/Zn cell under extremely high current density of 600 mA g⁻¹ even after 600th cycle with a capacity fading of only 0.0035% per cycle. This manuscript includes basic and application studies, and the results and discussions are well summarized and arranged systematically, showing new findings and valuable conclusion. Therefore, the reviewer basically recommends publication of this manuscript, but some revision and consideration should be required before publication.

Reviewer's comments are listed below.

The authors should describe the reason why the authors chose acetamide as solvent forming DES. This will be helpful information for the readers in this journal.

Results of thermo gravimetric analysis of ZES with molar ratio of 1:4, 1:5, and 1:9 should be added to supplementary file. This can be criterion to decide to choose ZES with 1:7 molar ratio.

Line 222

> This effective migration of the active Zn species could be attributed to the peculiar cationic structure of $[\text{ZnTFSA}(\text{acetamide})_n]^+$ with tethered anions (Fig. 2a inset), resulting in limited transport for negative charge carriers and underlying hopping-type ion transport mechanisms. ZES with molar ratio of 1:7 is a relatively dilute electrolyte. Even in such case, is it hopping-type ion transport ?

Fig. S11

The reviewer can observe oxidation current at around 2.4 V and 2.75 V. Which potential is involved in oxygen evolution ?

Line 217 (transference number)

It seems that the transference measurement does not base on the method shown in the reference. Therefore, the reviewer is wondering about comparison between their values. In addition, the method for transference number measurement should be explained in main text or supplementary file.

Line 232

> Interestingly, an overpotential of 0.185 V is required for the 1st cycle while roughly 0.1 V needed in the following cycles (2 to 10th) in ZES~
The reviewer thinks that the resulting low hysteresis (overvoltage) is simply due to the increase in surface area induced by morphology change. Is there any evidence to deny this?

Fig. S15

What is the reason of extremely low reversibility of Zn-deposition/stripping in 1 M $\text{Zn}(\text{TFSA})_2$? At the anodic process, what happens ? To begin with, why the reviewer choose 1 M $\text{Zn}(\text{TFSA})_2$ for comparison ?

Fig. 2d

The authors should display SEM images of Zn metal after cycling. The photo is not enough to demonstrate uniform Zn deposition in ZES.

Fig. 2f and Fig. S20

In Fig. S20 about 1 M Zn(TFSA)₂, it is clear semicircle associated charge transfer, whereas the reviewer cannot observe semicircle clearly. Why? Fitting plots should be overlapped.

Line 297

> It is also visible that the 295 Zn anode after deposition is covered by a thin surface layer (Fig. 3d inset), most likely in-situ formed SEI. The reviewer does not think so. Based only in the SEM observation result, it seems to overstatement.

The reviewer hopes these comments will be helpful.

Reviewer #2 (Remarks to the Author):

The manuscript describes the performance and mechanism of a deep eutectic solvent (DES) to enable reversible zinc electrodes for zinc ion batteries. The study is comprehensive with a suite of tools from computation to surface analysis. Electrochemical performance also appears to be promising. There are, however, several issues that concern the reviewer:

- 1) A more thorough discussion of the state of the art of zinc battery, including the use of DES, needs to be included. For example, this review contained specific references to the use of DES for zinc: <https://www.sciencedirect.com/science/article/pii/S2352152X17300476>. The readers would benefit from a comparison between this and other DES and better understand the uniqueness of the approach.
- 2) The core thesis, of the existence of the complex mono-anion, remains speculative. MassSpec might detect a fragment but that does not justify its existence under equilibrium states. That leaves the sole justification to be from DFT calculations.
- 3) The coulombic efficiency data need further clarification: In Figure 2c, data from 10 cycles are shown. What is the actual stable number and what might be the reasons for the non-ideal efficiency? The reviewer also found it difficult to consolidate this data with Figure 6f. How would 1.8x excess zinc produce 600 cycles given the efficiency?
- 4) The baseline system, 1M Zn(TFSI)₂, triggers a different electrode chemistry in the formation of ZnO. The comparison with the DES system does not have a clear scientific rationale. It would be more appropriate to compare with a concentrated pure zinc solution and how the differences in ionic structure determines the cycling performance.

In summary, the manuscript would make a good contribution to the field of zinc ion batteries by applying the SEI design method from other battery systems. It can be considered for publication if the concerns raised above are addressed.

Response to Reviewers

We would like to thank reviewers for their interest and time. The manuscript is revised according to their comments and suggestions. Below is a detailed description of the changes that have been made and our point to point response to the reviewers' comments.

Reviewer 1:

General Comments: This manuscript presents highly reversible Zn-deposition/stripping cycling by in-situ formed solid electrolyte interphase (SEI) originated from decomposition of complex of $[\text{Zn-TFSA}(\text{acetamide})_n]^+$ whose coordination environment was analyzed using Raman spectroscopy, FT-IR, and mass spectrometry. DES electrolyte consisted of $\text{Zn}(\text{TFSA})_2$ and acetamide with molar ratio of 1:7 dramatically improved cyclability of Zn electrode without dendrite morphology, whereas routine electrolyte of 1 M $\text{Zn}(\text{TFSA})_2$ showed poor cycling performance with tangle deposition morphology. The obtained high performance comes from ZnF_2 -rich SEI with mechanically rigidity and Zn^{2+} -permeability. The decomposition potential of the SEI was controlled by solvation state so that the SEI was formed at higher potential than that of Zn deposition. The authors achieved a reversible capacity of 51 mA h g⁻¹ in $\text{V}_2\text{O}_5/\text{Zn}$ cell under extremely high current density of 600 mA g⁻¹ even after 600th cycle with a capacity fading of only 0.0035% per cycle. This manuscript includes basic and application studies, and the results and discussions are well summarized and arranged systematically, showing new findings and valuable conclusion. Therefore, the reviewer basically recommends publication of this manuscript, but some revision and consideration should be required before publication.

Our Response: We are grateful to the reviewer for the positive comments and constructive suggestions as to how we could improve our manuscript.

Comment 1: The authors should describe the reason why the authors chose acetamide as solvent forming DES. This will be helpful information for the readers in this journal.

Our Response: The reviewer makes a good point. Acetamide has been demonstrated to be an effective ligand for developing zinc ion-conducting electrolytes (*J. Energy Storage* **2018**, *15*, 304; *Chem. Rev.* **2014**, *114*, 11060; *Electrochim. Acta* **2018**, *280*, 108), due to its good donor and acceptor abilities. Additionally, Acetamide-based DESs will be endowed with the advantages of relatively low viscosity and high ionic conductivity when acetamide serves as the HBD (*Chem. Soc. Rev.* **2012**, *41*, 7108), which has been revealed by our previous works (*Electrochim. Acta* **2018**, *280*, 108; *Nano Energy* **2019**, *57*, 625). The in-depth discussion of the reason why we chose acetamide as solvent forming DES, please see **Supplementary Notes for Fig. 1** in the revised supplementary information.

Supplementary information, page4: insert: "Supplementary Notes for Fig. 1:

Ace is a simple and typical donor molecule for deep eutectic solvents (DESs) due to its dipolar nature and has a strong ability to coordinate or H-bond with metal cations and anions by its two polar groups (C=O group and NH₂ group)⁶⁻⁹. The molten Ace also provides a high dielectric permittivity of 60 at 80 °C¹⁰. Its acid-base properties are similar to those of water,

and a variety of organic and inorganic compounds have been found to be soluble in Ace¹⁰. It is known that the physical/electrochemical properties of DESs are adaptable and dependent upon the donor moiety. DESs will be endowed with the advantages of relatively low viscosity and high ionic conductivity when Ace serves as the HBD¹¹, which has also been demonstrated by our previous works^{9,12}. Ace-based DESs have been reported as an alternative system for developing Zn electrolytes¹³. Additionally, amides are generally low-cost, chemically stable and environmentally friendly, which is beneficial for large-scale practical applications¹⁴.”

Comment 2: Results of thermo gravimetric analysis of ZES with molar ratio of 1:4, 1:5, and 1:9 should be added to supplementary file. This can be criterion to decide to choose ZES with 1:7 molar ratio.

Our Response: We thank the reviewer for the constructive suggestion. Results of thermo gravimetric analysis of ZESs with molar ratios of 1:4, 1:5, and 1:9 have been added to **supplementary Fig. 3** as suggested.

Page5: rephrase: “Furthermore, no phase change is observed in all ratios below 100 °C, and weight losses are only about 4.3% (1:9) and 3.3% (1:4) after heating at 100 °C (Supplementary Fig. 2, 3), reflecting the thermal adaptability of ZESs in the operating temperature region.”

Supplementary Fig. 3 Thermo gravimetric analysis (TGA) of ZESs with different Zn(TFSI)₂/Ace molar ratios.

Comment 3: Line 222

This effective migration of the active Zn species could be attributed to the peculiar cationic structure of [ZnTFSI(acetamide)_n]⁺ with tethered anions (Fig. 2a inset), resulting in limited transport for negative charge carriers and underlying hopping-type ion transport mechanisms. ZES with molar ratio of 1:7 is a relatively dilute electrolyte. Even in such case, is it hopping-type ion transport?

Our Response: This is a good question. Actually, the ZESs with the molar ratio ranging

from 1:4 to 1:9 seem not common dilute electrolytes. First, compared to the monovalent metal (Li or Na) salts, Zn salts with stronger anion-cation bonds are more difficult to be dissolved, which significantly limits the available metal concentration in electrolytes. Thus, it is more objective to compare the concentrations of different Zn electrolytes by the molar ratio of salt/solvent. If the acetamide is considered not only as a donor molecule but also a solvent, the molar ratios (1:4–1:9) of Zn(TFSI)₂/acetamide in ZESs are much higher than those of recently reported concentrated 3 M and 4 M Zn(TfO)₂ aqueous electrolytes (*J. Am. Chem. Soc.* **2016**, 138, 12894) (Zn/solvent molar ratios of 3 M and 4 M Zn(TfO)₂ aqueous electrolytes are 1:13.8 and 1:18.2, respectively). Second, the ZES with molar ratio of 1:7 can be classified into the “solvent-in-salt” electrolyte category, as the salt outnumbers the solvent in this system by weight (*Science* **2015**, 350, 938). In these solutions, the average number of solvent molecules available to solvate each ion is far below the “solvation numbers” that are established in conventional dilute electrolytes (*Nat. Commun.* **2013**, 4, 1481; *J. Phys. Chem. C* **2013**, 117, 4431). Instead, cation-anion interplays become more pronounced relative to solvent-ion interactions, allowing unusual physicochemical/interfacial properties. Third, due to the special formation mode of the ZES (strong intermolecular forces: hydrogen bonding and metal coordination) (*J. Energy Storage* **2018**, 15, 304; *Chem. Rev.* **2014**, 114, 11060), it features with an associated Zn²⁺-anion solvated structure (such as IIP or LIP, see **Fig. 1c**), which is analogous to the observations in concentrated electrolytes or locally concentrated electrolytes, essentially differing from that of dilute electrolytes with solvent-separated ions (*Nat. Energy* **2019**, 4, 269; *J. Phys. Chem. Lett.* **2016**, 7, 4795; *J. Phys. Chem. B* **2018**, 122, 2600). Thus, the present ZES with molar ratio of 1:7 can also be regarded as a unique concentrated electrolyte, and the ionic transport mechanism of it is expected to be different.

Considering that the associated cation-anion states have a close relationship with the hopping-type ion transport mechanism, it is thus speculated that the active Zn²⁺ species transport mode in the ZES might obey underlying hopping-type ion transport mechanisms. The fast ion transport ($1.66 \times 10^{-6} \text{ cm}^2 \text{ s}^{-1}$, **Supplementary Fig. 13**) and high active Zn²⁺ transport number (0.572, **Fig. 2a**) make this speculation reasonable (*Nat. Energy* **2019**, 4, 269-280). Of course, this theory refers to a new ion transmission mechanism proposed in recent years (*J. Phys. Chem. Lett.* **2016**, 7, 4795; *J. Phys. Chem. B* **2018**, 122, 2600; *Nat. Energy* **2019**, 4, 269), which needs further verification from both theoretical and experimental aspects. Based on this valuable comment, the manuscript has been revised accordingly.

p.10, Line 1 insert: “This effective migration of metal cations is most likely accounted for by the peculiar cationic Zn solvates with tethered anions⁴⁸ (Fig. 2a inset), and the resulting limited transport for negative charge carriers, which is analogous to the observations in highly concentrated electrolytes^{29,49}. Furthermore, the high Zn²⁺ transference number also implies that the ion transport manner in ZESs differs from those observed in the conventional dilute electrolytes; the active Zn²⁺ species might obey underlying hopping-type ion transport mechanisms (Zn²⁺ ions move from one anion to another through Lewis basic sites on TFSI⁻ with the aid of Ace matrix)^{29,49}.”

Comment 4: Fig. S11

The reviewer can observe oxidation current at around 2.4 V and 2.75 V. Which potential is involved in oxygen evolution?

Our Response: Thanks for this good question. Before the preparation of ZES electrolytes, the raw materials (acetamide and $\text{Zn}(\text{TFSI})_2$) were dried to remove residual water. The oxidation current here is independent of oxygen evolution from the water splitting, but attributed to the decomposition of electrolytes (acetamide or TFSI anion). In addition, the working electrode for testing the electrolyte electrochemical window also has an impact on the oxidation reactions (*Adv. Energy Mater.* **2017**, *7*, 1602055; *Energy Environ. Sci.* 10.1039/C9EE01699F). Based on a Ti electrode, the peak of the oxidation current at 2.4 V disappears. This result has been added in **Supplementary Fig. 16b**.

Supplementary Fig. 16 (a) Comparison of linear sweep voltammetry of ZES and 1 M $\text{Zn}(\text{TFSI})_2$ in Zn/SS cells at the scanning rate of 1 mV/s. The working and counter electrodes are Ti and Zn, respectively. Insets: the optical photographs of Zn anodes and separators after testing in (I) ZES and (II) 1 M $\text{Zn}(\text{TFSI})_2$. (b) Linear sweep voltammetry of ZES at the scanning rate of 1 mV s^{-1} with SS and Ti as working electrodes, respectively.

Supplementary Notes for Fig. 16, rephrase: “Before the preparation of ZESs, the raw materials (Ace and $\text{Zn}(\text{TFSI})_2$) were dried to remove residual water. The oxidation current of ZES is decoupled from the O_2 -evolution reaction, but attributed to the decomposition of electrolyte components. In addition, the working electrode for testing the electrolyte

electrochemical window has an impact on the oxidation reactions (Supplementary Fig. 16b)¹⁷.”

Comment 5: Line 217 (transference number)

It seems that the transference measurement does not base on the method shown in the reference. Therefore, the reviewer is wondering about comparison between their values. In addition, the method for transference number measurement should be explained in main text or supplementary file.

Our Response: We thank the reviewer for carefully reviewing our manuscript and raising this point. In the references, the moving boundary method is adopted for the measurement of transference number of Zn²⁺. The test objects of this method are mostly aqueous solutions (*Chem. Rev.* **1932**, 11, 171). However, the operation process is relatively complicated, and the used cadmium electrode has certain toxicity. Thus, this method is rarely applied in recent investigations. The method for measuring the transference number of ions used in our work is reliable and widely acknowledged (*Energy Environ. Sci.* **2019**, 12, 1938; *Nat. commun.* **2013**, 4, 1481; *Chem. Mater.* **2013**, 25, 834; *J. Phys. Chem. C* **2016**, 120, 4276). The details for measuring Zn²⁺ transference number were supplemented in the **Experimental Part in Supplementary information**.

Experimental Part in Supplementary information: insert: “Zn²⁺ transference number was evaluated in symmetrical Zn battery combined by EIS before and after the chronoamperometry (CA) test, and calculated by the following equation [2]:

$$T = \frac{I_s(\Delta V - I_0 R_0)}{I_0(\Delta V - I_s R_s)} \quad [2]$$

where ΔV is the voltage polarization applied, I_s and R_s are the steady state current and resistance, respectively, and I_0 and R_0 are the initial current and resistance, respectively. The applied voltage polarization here is 5 mV.”

Comment 6: Line 232

Interestingly, an overpotential of 0.185 V is required for the 1st cycle while roughly 0.1 V needed in the following cycles (2 to 10th) in ZES. The reviewer thinks that the resulting low hysteresis (overvoltage) is simply due to the increase in surface area induced by morphology change. Is there any evidence to deny this?

Our Response: This is a good suggestion to correct the controversial statement in our work. We agree with the reviewer that the resulting low hysteresis (overvoltage) is due to the increase in surface area induced by morphology change. From the observation of Ti matrix after cycling, it was found that specific surface area of Ti increased (**Supplementary Fig. 20c**). Note that the stepwise generation of the SEI also causes a change in surface area. In addition, there is a possibility that the SEI may be broken and then repaired locally during Zn plating/stripping cycling with micron-sized volumetric change, which also incurs a change in the surface area of the electrode (*Energy Environ. Sci.* 2017, 10, 580; *J.*

Electrochem. Soc. 2017 164, A2418). Meanwhile, this phenomena can also be observed in previous works (*Angew. Chem. Int. Ed.* **2016**, 55, 2889; *Adv. Mater.* **2018**, 30, 1706102), which was attributed to the SEI formation during initial cycling and subsequent stabilization. To avoid misunderstanding, we have changed the description in the revised manuscript, as shown below.

Page10, line19: rephrase: “Interestingly, an overpotential of 0.185 V is required for the 1st cycle in ZES while roughly 0.1 V needed in the following cycles (Fig. 2b, green circle), which suggests the increase in surface area as well as the progressively improved stability induced by stepwise generation of the in-situ formed interphase^{55,56}.”

Supplementary Fig. 20 (a) CEs of Zn plating/stripping versus cycle number plot for Zn/Ti cells cycled at 0.5 mA cm⁻² and a deposition capacity of 1 mAh cm⁻². (b) Galvanostatic Zn plating/stripping at 0.5 mA cm⁻² in 1 M Zn(TFSI)₂. The working and counter electrodes are Ti and Zn, respectively. SEM images of Zn deposits on Ti after 10 cycles of galvanostatic Zn plating/stripping at a 0.5 mA cm⁻² (1 mAh cm⁻²) in (c) ZES and (d) 1 M Zn(TFSI)₂. The working and counter electrodes are Ti and Zn, respectively.

Comment 7: Fig. S15

What is the reason of extremely low reversibility of Zn-deposition/stripping in 1 M Zn(TFSA)₂? At the anodic process, what happens? To begin with, why the reviewer choose 1 M Zn(TFSA)₂ for comparison?

Our Response: We thank the reviewer for carefully reviewing our manuscript and for raising these questions.

1) The low reversibility of Zn-deposition/stripping in 1 M Zn(TFSI)₂ is mainly due to uncontrolled dendrites (**Fig. S20d**) and notorious side-reactions (H₂ evolution, passivation and corrosion) occurred at the Zn-electrolyte interface (*Adv. Energy Mater.* **2018**, 8,

1702097; *Nano Energy* **2019**, *57*, 625), resulting in a large amount of irreversible Zn consumption (electrochemically inactive products or dead Zn; e.g., the formation of water-insoluble ZnO and $x\text{ZnCO}_3 \cdot y\text{Zn(OH)}_2 \cdot z\text{H}_2\text{O}$, etc. is shown in **Fig. S25**). Moreover, Zn loss also includes the reaction loss associated with the side reactions between Zn and deposition substrate surface (*Adv. Energy Mater.* **2018**, *8*, 1702097). Notably, there is no consensus on suitable Zn deposition substrates for evaluating the Zn electrolytes. Stainless steel (SS), Ti and Mo have been applied in previous works (*Energy Environ. Sci.* **2019**, *12*, 1938; *Chem. Commun.* **2018**, *54*, 14097; *Adv. Energy Mater.* **2019**, 1900196). The unsatisfactory compatibility between Ti substrate and the 1 M Zn(TFSI)₂ is also responsible for the poor reversibility.

2) At the anodic process, as is shown in **Supplementary Fig. 20**, side-reactions (such as competitive H₂ evolution reaction) remain, and consume charge and freshly deposited Zn irreversibly. Meanwhile, the H₂ evolution reaction further elevates the pH at the electrolyte-electrolyte interfacial region. The corresponding electronically insulating byproducts formed in deposition process block the charge transfer, further aggravating the irreversibility. Similar observations on aqueous electrolytes can also be obtained in previous works (*Nat. Chem.* **2018**, *10*, 532; *Nat. Mater.* **2018**, *17*, 543; *Nat. Energy* **2017**, *2*, 17119; *J. Am. Chem. Soc.* **2016**, *138*, 12894) and our recent analysis (*Nano Energy* **2019**, *57*, 625).

3) The reason why we choose 1 M Zn(TFSI)₂ for comparison is as follows. Currently, there are still no reliable and recognized electrolytes to achieve good reversibility of Zn deposition/stripping both from electrochemical and thermodynamic aspects (*Nano Energy* **2019**, *57*, 625). As such, recent works on exploring new Zn electrolytes use alkaline and neutral aqueous electrolytes for comparison, but several issues, in particular severe dendrite growth during cycling and metal passivation (*Nat. Mater.* **2018**, *17*, 543; *J. Energy Storage* **2018**, *15*, 304; *Chem. Commun.* **2018**, *54*, 14097; *Adv. Energy Mater.* **2019**, *9*, 1900196) remain. More importantly, in order to ensure the consistency between samples in terms of anion and cation, Zn(TFSI)₂ is also applied in the control electrolyte group. However, due to the high charge density of Zn²⁺, Zn(TFSI)₂ is difficult to dissociate in common organic solvent systems. Thus, water is used as the solvent since it can make Zn(TFSI)₂ more easily dissociated compared with acetonitrile or ionic liquids. Here, taking into account the pH, the ionic conductivity as well as the electrochemical performance (**Supplementary Fig. 14, 15**) of different concentrations, we screened 1 M Zn(TFSI)₂ as the control group (**Supplementary Fig. 14b**). Although the Zn(TFSI)₂ concentration can reach to 1.5 M based on water as the solvent, the stronger acidity will bring about the corrosion of Zn anode and current collectors, and sustained consumption of water, resulting in poor long-term stability (*Angew. Chem. Int. Ed.* **2016**, *55*, 2889; *Chem. Soc. Rev.* **2014**, *43*, 5257; *Adv. Funct. Mater.* **2018**, *28*, 1802564).

Moreover, taking into account the proposed anion-derived SEI formation, various Ace-based DESs have been prepared based on various common Zn salts (Zn(ClO₄)₂, Zn(CH₃COO)₂, Zn(BF₄)₂ and Zn(TFSI)₂), and thus the different ion species can be obtained. From the comparison in terms of the cycling performance for Zn plating/stripping and the oxidation stability (**Supplementary Fig. 17 and 36**), it is apparent that ZES indeed exhibits lower polarization and better stability compared with DESs based on other anions,

demonstrating the TFSI⁻-induced SEI formation mechanism and the uniqueness of the SEI composition.

Based on this suggestive comment, our manuscript has been revised accordingly. Corresponding comparing results have been added in the **Supplementary Information**.

Page8, line30: rephrase: “Taking the physical/chemical properties and cost factors into consideration, we chose the molar ratio of 1:7 as the main research object (for the selection of the control group see Supplementary Fig. 14, 15).”

Page10, line15: rephrase: “Of note, the CE of the first 10 cycles in ZES rises gradually to above 98.0%; instead, the inferior CE of less than 70% was obtained in 1 M Zn(TFSI)₂ (Supplementary Fig. 20a, b) under identical conditions, which could be ascribed to the severe parasitic reactions that simultaneously occurred during Zn deposition¹⁷, along with uncontrolled dendrites (Supplementary Fig. 20d)^{12,54}.”

Page17, line21: insert: “Note that ZES also exhibits lower polarization and better stability for Zn/Zn²⁺ reactions compared with DESs based on other Zn salts (Supplementary Fig. 36), demonstrating the TFSI⁻-induced SEI formation mechanism and the uniqueness of the SEI composition.”

Supplementary information, page16: insert: “Supplementary Notes for Fig. 14:

In order to ensure the consistency between samples in terms of anion and cation, Zn(TFSI)₂ is also applied in the control electrolyte group. However, due to the high charge density of Zn²⁺, Zn(TFSI)₂ is difficult to dissociate in organic solvents. Thus, water is used as the solvent since it can make Zn(TFSI)₂ more easily dissociated, though the Zn salt will be precipitated in the 2 M Zn(TFSI)₂ after resting for 24 hours, as shown in Supplementary Fig. 14a. Here, taking into account the pH, the ionic conductivity as well as the electrochemical performance (Supplementary Fig. 15) of different concentrations, we choose 1 M Zn(TFSI)₂ as the control sample (Supplementary Fig. 14b).”

Supplementary information, page17: insert: “Supplementary Notes for Fig. 15:

1 M Zn(TFSI)₂ exhibits extended cycle life and lower polarization at both low (0.1 mA cm⁻²) and high (1 mA cm⁻²) rates compared with 1.5 M Zn(TFSI)₂.”

Supplementary Fig. 14 (a) Images of Zn(TFSI)₂ aqueous electrolytes (resting for 24 hours). (b) Variation of pH and conductivity of Zn(TFSI)₂ aqueous electrolytes with different concentrations.

Supplementary Fig. 15 Voltage responses of Zn/Zn symmetric cells in 1.5 M and 1 M Zn(TFSI)₂ electrolytes at (a) 0.1 mA cm⁻² (0.05 mAh cm⁻² for each half cycle) and (b) 1 mA cm⁻² (0.5 mAh cm⁻² for each half cycle), respectively.

Supplementary Fig. 36 Voltage responses of Zn/Zn symmetric cells in DES electrolytes formed by different Zn salts ($\text{Zn}(\text{ClO}_4)_2$, $\text{Zn}(\text{CH}_3\text{COO})_2$ and $\text{Zn}(\text{TFSI})_2$) at 0.01 mA cm^{-2} (0.5 h for each half cycle).

Comment 8: Fig. 2d

The authors should display SEM images of Zn metal after cycling. The photo is not enough to demonstrate uniform Zn deposition in ZES.

Our Response: We appreciate this helpful suggestion. SEM images of Zn metal after cycling have been added to **Supplementary Fig. 22**. The results are consistent with the optical images. The surface SEM image of cycled Zn in ZES is visually uniform (**Supplementary Fig. 22c**), while characteristic Zn protrusions are shown in the case using 1 M $\text{Zn}(\text{TFSI})_2$ (**Supplementary Fig. 22b**).

Page11, line14: rephrase: “Note that the surface morphology of cycled Zn in ZES is visually uniform (Fig. 2d inset right and Supplementary Fig. 22c), while characteristic Zn protrusions are shown in the case using 1 M $\text{Zn}(\text{TFSI})_2$ (Fig. 2d inset left and Supplementary Fig. 22b).”

Supplementary Fig. 22 (a) Voltage responses of the Zn/Zn symmetric cell under repeated polarization in ZES at 0.1 mA cm^{-2} (before the 1,000th cycle) and 0.5 mA cm^{-2} (after the 1,000th cycle). SEM images of the cycled Zn after (b) 180 cycles in 1 M $\text{Zn}(\text{TFSI})_2$ and (c) 2,000 cycles in ZES at a rate of 0.1 mA cm^{-2} (0.05 mAh cm^{-2}).

Comment 9: Fig. 2f and Fig. S20

In Fig. S20 about 1 M Zn(TFSA)₂, it is clear semicircle associated charge transfer, whereas the reviewer cannot observe semicircle clearly. Why? Fitting plots should be overlapped.

Our Response: Thanks for this helpful suggestion. Fig. 2f and Fig. S20 are adjusted to Fig. S24. The obvious semicircle associated with charge transfer in 1 M Zn(TFSI)₂ can be observed after the refitting operation. Corresponding EIS results have been added in Supplementary Fig. 24d of the revised manuscript, as suggested.

Supplementary Fig. 24 EIS data measured with Zn/Zn symmetric cells in (a) ZES and (b) 1 M Zn(TFSI)₂ at different galvanostatic cycles at 0.1 mA cm⁻². Scatter plots and dashed lines denote experimental spectra and fitting curves of impedance, respectively. (c) The evolution of the charge-transfer resistance in different electrolytes. Insets in (a) and (b) exhibit the equivalent circuit model of EIS. R_b stands for the electrolyte bulk resistance, R_{inter} and CPE₁ are the interface resistance and its related double-layer capacitance, which correspond to the semicircle at high frequencies, R_{ct} and CPE₂ represent the charge transfer resistance and its related double-layer capacitance, which correspond to the semicircle at medium frequencies, and W_o represents the Warburg impedance related the diffusion of Zn-ions, which is indicated at low frequencies²⁵.

Comment 10: Line 297

It is also visible that the 295 Zn anode after deposition is covered by a thin surface layer (Fig. 3d inset), most likely in-situ formed SEI. The reviewer does not think so. Based only in the SEM observation result, it seems to overstatement.

Our Response: This is a good suggestion to correct the controversial statement in our work. We agree with the reviewer. To avoid any sort of controversy, we have modified the description of this place in the revised manuscript.

Page13, line7: rephrase: “It is also visible that the Zn anode after deposition is covered by a thin surface layer (Fig. 3d inset), also corresponding to the surface modification. Thus, it is reasonable to assume that this additional Zn-electrolyte interphase dictates the reversible Zn/Zn²⁺ redox with efficient Zn²⁺ transport and deposition (Fig. 3b).”

Reviewer 2:

The manuscript describes the performance and mechanism of a deep eutectic solvent (DES) to enable reversible zinc electrodes for zinc ion batteries. The study is comprehensive with a suite of tools from computation to surface analysis. Electrochemical performance also appears to be promising. There are, however, several issues that concern the reviewer:

Our Response: We thank the reviewer for the encouraging comments.

Comment 1: A more thorough discussion of the state of the art of zinc battery, including the use of DES, needs to be included. For example, this review contained specific references to the use of DES for zinc: <https://www.sciencedirect.com/science/article/pii/S2352152X17300476>. The readers would benefit from a comparison between this and other DES and better understand the uniqueness of the approach.

Our Response: We appreciate the suggestion of the reviewer. This review is good reference for our work and very helpful for the introduction of DES electrolytes, which have been cited in the revised manuscript (**Ref 26**) and supplementary information (**Ref 13**). Correspondingly, a more thorough discussion has been added in the revised manuscript.

Page3, line20: insert: “As a new class of versatile fluid materials, the deep eutectic solvents (DESs), generally created from eutectic mixtures of Lewis or Brønsted acids and bases that can associate with each other, have been found to be interesting on account of their excellent dissolution ability, even for the multivalent metal salts and oxides^{26,27}.”

Supplementary Notes for Fig. 1, insert: “Ace-based DESs have been reported as an alternative system for developing Zn electrolytes¹³. Additionally, amides are generally low-cost, chemically stable and environmentally friendly, which is beneficial for large-scale practical applications¹⁴.”

Comment 2: The core thesis, of the existence of the complex mono-anion, remains speculative. MassSpec might detect a fragment but that does not justify its existence under equilibrium states. That leaves the sole justification to be from DFT calculations.

Our Response: Thank the reviewer for carefully reviewing our manuscript and raising this very valuable advice. Combining various spectroscopic and computational techniques, we have examined the solution structure of ZESs more rigorously. Finally it is concluded that in 1:7 and 1:9 solutions, the majority of the dominant monomeric Zn species are coordinated by a single TFSI⁻ anion, while the fraction of neutral Zn complexes coordinated by two TFSI⁻ anions increases for higher salt contents (1:4 and 1:5 solutions), whereas no three TFSI⁻ coordination case was found.

In greater detail, from the Raman and FTIR analysis, it is apparent that there is an unequivocal ionic association relationship between Zn²⁺ and TFSI⁻ (shown in **Fig. 1a, b of revised manuscript**). A deconvolution analysis of Raman vibration mode (S–N–S) of TFSI⁻ shows that in 1:7 and 1:9 solutions, the majority of TFSI⁻ anions exist as long-lived loose ion pairs (LIPs), while the cation-anion interactions become intensified with more intimate

ion pairs (IIPs) (*J. Electrochem. Soc.* **2012**, 159, A1489; *J. Am. Chem. Soc.* **2017**, 139, 18670) formed as salt concentration increases (1:4 and 1:5 solutions) (**Fig. 1d**). This dependence of the ionic association on the salt/Ace ratios can be clearly reflected by HRMS (**Supplementary Fig. 7**). Subsequently, the HRMS signals of ZESs also corroborate the existence of LIPs and IIPs in all given ratios, further indicating Zn^{2+} is more likely to coordinate with a single TFSI^- . It should be noted that the only anionic species of TFSI^- found in HRMS suggests a low possibility of anionic Zn complexes associated with more than two TFSI^- anions (**Supplementary Fig. 7**).

Theoretical simulations further identify that for the 1:7 ratio, on average, one TFSI^- anion could be observed in each Zn^{2+} primary solvation sheath, typically in the form of the $[\text{ZnTFSI}(\text{Ace})_2]^+$ solvate (**Supplementary Fig. 8a, b**); as expected, in 1:4 ZES, more TFSI^- anions enter the Zn^{2+} solvation sheath (**Supplementary Fig. 8c, d**) with an increased fraction of neutral Zn complexes coordinated by two TFSI^- anions. However, no three TFSI^- coordination case was found (**Supplementary Fig. 8d**). Due to the structural flexibility and extensive charge delocalization, the bulky TFSI^- anion may be coordinated in varying ways to Zn^{2+} cations (*J. Electrochem. Soc.* **2012**, 159, A1489-A1500), incurring dynamic equilibria of cationic or neutral species with various conformations (as shown in **Fig. 1g**). The density functional theory (DFT) geometry optimization of $[\text{ZnTFSI}(\text{Ace})_n]^+$ complexes predicts that the $[\text{ZnTFSI}(\text{Ace})_2]^+$ structure with bidentate coordination by TFSI^- possesses the most uniform molecular electrostatic potential energy surface distribution along with relatively low total binding energy (**Fig. 1f and Supplementary Fig. 9, 11, 12**), in line with the predominant signal detected from HRMS. However, a lower tendency of bidentate coordination was shown in the neutral $[\text{ZnTFSI}_2(\text{Ace})_n]$ complexes (**Supplementary Fig. 10**), which is attributed to the steric-hindrance effect. Despite the discrepancies between the simulation and experimental results, it is quite informative to examine the solution structure of ZESs. Accordingly, more detailed analysis and discussion of electrolyte solvation structure have been added in the revised manuscript, as shown below.

Page7, line3: insert: “Essentially, the latter is identical to that in crystal lattice (741.6 cm^{-1}), indicative of a possible pronounced interionic attraction in ZESs⁴³. Turning to the Raman vibration mode of TFSI^- at the same region (**Fig. 1c**), a deconvolution analysis shows that the Raman band consists of three modes at 740 , 744 , and $748/747\text{ cm}^{-1}$, arising from free anions (FA)-($\#\text{Zn}^{2+} = 0$), loose ion pairs (LIP)-($\#\text{Zn}^{2+} = 1$) and intimate ion pairs (IIP)-($\#\text{Zn}^{2+} = 1$), respectively^{22,37,45}. In all cases of the ZES system, albeit without obvious ionic aggregates (AGG; the anions are coordinated to two or more cations)³⁷, the ubiquitous presence of cation-anion coordination can be identified. In 1:7 and 1:9 solutions, the majority of TFSI^- anions exist as long-lived LIPs, suggesting the dominant monomeric Zn species coordinated by TFSI^- , while the ionic association becomes stronger with more IIPs formed at relatively higher salt contents (1:4 and 1:5 solutions) (**Fig. 1d**). Effects related to the salt concentration are also imposed on the drastic variation in viscosity and ion conductivity (**Supplementary Fig. 13c**).

The high-resolution mass spectra (HRMS) of ZESs testify the existence of LIPs and IIPs in all given ratios. Typically, distinct signals of various cationic TFSI^- -containing complexes ($[\text{ZnTFSI}(\text{Ace})]^+$ at $m/z=403$, $[\text{ZnTFSI}(\text{Ace})_2]^+$ at $m/z=462$, and $[\text{ZnTFSI}(\text{Ace})_3]^+$ at $m/z=521$) can be detected, but without evidence of free Zn^{2+} ions (**Supplementary Fig. 6**).

Moreover, the variation trend of these cationic peak intensities qualitatively indicates a more pronounced ionic association upon increasing the Zn-salt content, in line with the above Raman results (Fig. 1c, d). It should be noted that the only anionic species of TFSI⁻ found in HRMS suggests a low possibility of anionic Zn complexes (monomeric) with more than two associated TFSI⁻ anions (Supplementary Fig. 7).

Theoretical simulations were performed to further identify the ion speciation of ZESs. In both cases of mixtures (1:7 and 1:4), molecular dynamics (MD) simulations predict a competition between the Ace and TFSI⁻ for coordination to Zn²⁺ cations (Supplementary Fig. 8). For the 1:7 ratio, one TFSI⁻ anion (on average) could be observed in each Zn²⁺ primary solvation sheath, typically in the form of the [ZnTFSI(Ace)₂]⁺ solvate (Supplementary Fig. 8a, b). However, in 1:4 ZES, where only four Ace molecules per Zn(TFSI)₂ are involved in eutectic solution formation, a lower Ace population is available for Zn²⁺ solvation and H-bonding with TFSI⁻ anions simultaneously⁴²; instead, more TFSI⁻ anions enter the Zn²⁺ solvation sheath (Supplementary Fig. 8c, d). The fraction of neutral Zn complexes coordinated by two TFSI⁻ anions is thus expected to increase, whereas no three TFSI⁻ coordination case was found (Supplementary Fig. 8d). Apparently, ZES is a system featured with the existence of anion-associated Zn solvates, and the ionic interplay strength can be tuned through simple regulation of the Zn(TFSI)₂/Ace ratio. By virtue of the structural flexibility, the TFSI⁻ anion may be coordinated in varying ways to Zn²⁺ cations³⁷, incurring dynamic equilibria of cationic or neutral species with various conformations (Fig. 1g).

Given the fact that TFSI⁻ is more likely to form bidentate coordination to a single cation than other common anions (*i.e.*, PF₄⁻, ClO₄⁻ and BF₄⁻)⁴⁶, the local atomic configurations of Zn complexes were investigated theoretically. The density functional theory (DFT) geometry optimization of [ZnTFSI(Ace)_n]⁺ complexes verifies the preference of the C=O group of Ace and both two O atoms of TFSI⁻ for the coordination with the central Zn²⁺ cation (Supplementary Fig. 9, 11). The [ZnTFSI(Ace)₂]⁺ structure with bidentate coordination by TFSI⁻ possesses the most uniform molecular electrostatic potential energy surface distribution along with relatively low total binding energy (Fig. 1f and Supplementary Fig. 9, 11, 12), in reasonable agreement with the predominant signal of cationic species observed from HRMS. Note that the steric-hindrance effect caused by the bulky TFSI⁻ also dictates the identity of the solution species. This can be reflected by the absence of anionic Zn solvates and a lower tendency of bidentate coordination in [ZnTFSI₂(Ace)_n] complexes (Supplementary Fig. 10).”

Fig. 1 Structure analysis of ZESs and identity of the ionic species. **a)** Raman, **b)** FTIR and **c)** Fitted Raman spectra of ZESs with different Zn(TFSI)₂/Ace molar ratios (1:9–1:4). Solid and dashed lines denote experimental spectra and fitting curves, respectively. **d)** Solvate species distribution in ZESs (free anions (FA), loose ion pairs (LIP) and intimate ion pairs (IIP)), all obtained from the fitted Raman spectra. **e)** Schematic diagram of the interplay among Zn²⁺, TFSI⁻ and Ace to form eutectic solutions. **f)** Molecular electrostatic potential energy surface of [ZnTFSI(Ace)₂]⁺ (C₂-O-II, bidentate coordination of TFSI⁻) based on density functional theory (DFT) simulation. Electron density from total self-consistent-field (SCF) density (isoval = 0.001). **g)** Illustration of representative environment of active Zn species within the ZES.

Supplementary Fig. 8 MD studies of the Zn²⁺-solvation structure. Snapshots of the MD simulation boxes for the ZESs with Zn(TFSI)₂/Ace molar ratios of **(a)** 1:7 and **(c)** 1:4 at 353 K. Representative Zn²⁺-solvation structures for **(b)** 1:7 and **(d)** 1:4.

Supplementary Fig. 9 Optimized geometries of (a) $[\text{ZnTFSI}(\text{Ace})]^+$ (C_1 -O-II), (b) $[\text{ZnTFSI}(\text{Ace})_2]^+$ (C_1 -O, N-I), (c) $[\text{ZnTFSI}(\text{Ace})_2]^+$ (C_1 -O-II), (d) $[\text{ZnTFSI}(\text{Ace})_2]^+$ (C_2 -O-II), (e) $[\text{ZnTFSI}(\text{Ace})_3]^+$ (C_1 -O-II), (f) $[\text{ZnTFSI}(\text{Ace})_4]^+$ (C_1 -O-I) based on DFT calculations (Zn-purple, N-blue, O-red, S-yellow, F-light blue, C-light gray, H-white). E : the total binding energy between Zn^{2+} , TFSI^- and Ace. The TFSI^- anion is known to have two different low-energy conformational states: a *cisoid* form (C_1) with the CF_3 groups on the same side of the S-N-S plane and a *transoid* form (C_2) with the CF_3 groups on opposite sides of the plane¹⁷. I, monodentate coordinated TFSI^- anions; II, bidentate coordinated TFSI^- anions.

Supplementary Notes for Fig. 9:

The DFT geometry optimization of the $[\text{ZnTFSI}(\text{Ace})_4]^+$ complex started from a configuration with four Ace molecules bound to a Zn^{2+} cation, but the optimized geometry instead converging to a configuration with only three of the Ace molecules bound to the Zn^{2+} cation while the other Ace is directed away from the central cation (Supplementary Fig. 9f).

Supplementary Fig. 10 Optimized geometries of the (a) $[\text{ZnTFSI}_2(\text{Ace})_2]$ (C_1 -O-I, C_1 -O-II), (b) $[\text{ZnTFSI}_2(\text{Ace})_2]$ (C_1 -O-II, C_2 -O-I), (c) $[\text{ZnTFSI}_2(\text{Ace})_2]$ (C_1 -O-I, C_2 -O-II), (d) $[\text{ZnTFSI}_2(\text{Ace})_2]$ (C_1 -O-I, C_1 -O-I), (e) $[\text{ZnTFSI}_2(\text{Ace})_2]$ (C_1 -O-II, C_2 -O-II), (f) $[\text{ZnTFSI}_2(\text{Ace})_3]$ (C_1 -O-II, C_2 -O-I), (g) $[\text{ZnTFSI}_2(\text{Ace})_3]$ (C_1 -O-II, C_2 -O-I), (h) $[\text{ZnTFSI}_2(\text{Ace})_4]$ (C_1 -O-I, C_1 -O-I).

[ZnTFSI₂(Ace)₃] (C₂-O-I, C₂-O-I), (h) [ZnTFSI₂(Ace)₄] (C₁-O-I, C₂-O-I) based on DFT calculations (Zn-purple, N-blue, O-red, S-yellow, F-light blue, C-light gray, H-white).

Supplementary Notes for Fig. 10:

The DFT geometry optimization of the [ZnTFSI₂(Ace)₄] complex started from a configuration with four Ace molecules bound to a Zn²⁺ cation, but the optimized geometry is similar to the [ZnTFSI(Ace)₄]⁺ case.

Supplementary Fig. 11 The total interaction energy (E) of [ZnTFSI(Ace)₂]⁺ with three different geometries ((C₂-O-II), (C₁-O-I, C₁-N-I) and (C₁-O-II), respectively) (Zn-purple, N-blue, O-red, S-yellow, F-light blue, C-light gray, H-white).

Supplementary Fig. 12 The molecular electrostatic potential energy surface of [ZnTFSI(Ace)_n]⁺ ($n = 2$ or 3) species based on DFT. Electron density from total SCF density (isoval = 0.001).

Comment 3: The coulombic efficiency data need further clarification: In Figure 2c, data from 10 cycles are shown. What is the actual stable number and what might be the reasons for the non-ideal efficiency? The reviewer also found it difficult to consolidate this data with Figure 6f. How would 1.8 x excess zinc produce 600 cycles given the efficiency?

Our Response: Many thanks to this comment. The actual stable number exceeded 200 cycles using Zn/Ti cell at an average deposition capacity of $\sim 0.61 \text{ mAh cm}^{-2}$ (**Supplementary Fig. 21**).

1) Firstly, it should be pointed out here that the SEI formation in ZES is stepwise, as evidenced by the gradual stabilization process of the Zn-anode potential with cycling (**Fig. 2d**). In addition to evaluate the CE, **Fig. 2b** demonstrated the interfacial change upon continuous Zn plating/stripping in ZES. This SEI layer on Zn anode contains a large number of components, such as the typical ZnF_2 phase (**shown in Fig. 4**), which contributes to the main consumption of Zn and thereby the non-ideal CE in the initial stage. In the following cycling steps, there is a possibility that the SEI may be broken and then self-healing, further consuming active Zn slightly (*Energy Environ. Sci.* **2017**, 10, 580; *J. Electrochem. Soc.* **2017** 164, A2418). Subsequently, the SEI layer was stabilized, along with the gradual improvement of CE (see **Supplementary Fig. 20, 21**). As a reliable method (*Nat. Mater.* **2018**, 17, 543; *J. Am. Chem. Soc.* **2016**, 138, 12894; *Energy Environ. Sci.* 10.1039/C9EE01699F), cyclic voltammetry (CV) has been further applied to evaluate CE of ZES using Zn/Ti cell at an average deposition capacity of $\sim 0.61 \text{ mAh cm}^{-2}$ (**Supplementary Fig. 21**). Corresponding chronocoulometry curves (**Fig. 2c**) reveal that the plating/stripping is highly reversible with CE approaching 100% after the initial 30 conditioning cycles (an average CE of **99.7%** for 200 cycles). From this aspect, ZES is superior to other reported concentrated Zn electrolytes (30 m ZnCl_2 , *Chem. Commun.* 2018, 54, 14097; $\text{ZnCl}_2 \cdot 2.3\text{H}_2\text{O}$, *Adv. Energy Mater.* **2019**, 1900196; 2 M ZnSO_4 and 2 M $\text{Zn}(\text{CH}_3\text{COO})_2$, *Nat. Mater.* **2018**, 17, 543), with CEs of 95.4%, 98.7%, 75% and 80%, respectively (see **Supplementary Table 4**). The realization of secondary Zn-metal cells with a long cycle (or calendar) life can be thus anticipated, especially when the excess of Zn anode is limited.

Secondly, the non-ideal CE does not necessarily refer to an irreversible Zn consumption, but is also caused by the poor compatibility between electrolyte and deposition substrate (*Adv. Energy Mater.* **2018**, 8, 1702097). The CE evaluation of Zn plating/stripping has a great dependence on the deposition substrate (inert electrode), concerning their intrinsic physicochemical properties as well as the surface roughness and/or treatment conditions. We have confirmed this through the comparative experiments. As shown in **Supplementary Fig. 19**, for different substrates (Ti, SS, Mo, etc.), Zn plating/stripping efficiency varies significantly. Moreover, during the initial Zn plating process, Zn loss not only includes the side products formed by electrolyte decomposition, but also involves the side reactions associated with the substrate surface (*Adv. Energy Mater.* **2018**, 8, 1702097; *Energy Environ. Sci.* 10.1039/C9EE01699F). The deposition substrates that are universally applicable to evaluating various Zn electrolyte systems are worth studying in the following works.

2) In the present work, the “1.8 × excess Zn” is calculated based on the **theoretical** capacity of V_2O_5 (290 mAh g^{-1}) as shown in the experimental section. However, the

Zn/V₂O₅ battery cycled at a high rate of 600 mA/g practically exerts a capacity of about 48 mAh g⁻¹ (based on the active V₂O₅) with an areal capacity of 0.7 mAh cm⁻², ~1/6 of the theoretical capacity. Therefore, from the testing aspect, the Zn excess (14.28 mg cm⁻², 11.7 mAh cm⁻²) is 15.7 times. We very appreciate the reviewer for spotting this oversight. To avoid misunderstanding, we have changed the description of “the Zn excess” in the revised manuscript, as shown below.

Obviously, the average CE of the Zn/V₂O₅ cells approaches 99.9%, which is greater than the CE obtained by the Zn/Ti cell (98.0% and 99.7%, evaluating by galvanostatic and CV method, respectively). Note that, even with CE of 98.0% for estimation, the 15.7 × excess Zn can fully support the normal operation of the battery over 600 cycles.

The capacity loss per cycle: 0.7 mAh cm⁻² × 0.02 = 0.014 mAh cm⁻²; the total capacity loss after 600 cycles: 0.014 mAh cm⁻² × 600 = 8.4 mAh cm⁻².

Furthermore, given 16.7 times Zn (15.7 × excess) at anode based on the capacity of active V₂O₅, we assumed that only one time of Zn remains after cycling, and then calculated the cycle number that 15.7 times of Zn (i.e., the excess part) could support under different CE conditions (*Prog. Mater. Sci.* **2017**, 89, 479). The complete consumption of Zn is defined as 1% of Zn retention. As shown below (*n* is the cycle number that one time Zn could support), even if all 15.7 × excess Zn was consumed, the normal operation of the cell can be guaranteed.

$98.0\%^n = 1\%$;	$99.7\%^n = 1\%$	$99.9\%^n = 1\%$;
$\log 98.0\%^n = \log 1\%$;	$\log 99.7\%^n = \log 1\%$;	$\log 99.9\%^n = \log 1\%$;
$n \approx 228$;	$n \approx 1,532$;	$n \approx 4,602$;
$228 \times 15.7 \approx 3,579$.	$1532 \times 15.7 \approx 24,052$.	$4602 \times 15.7 \approx 72,251$.

Furthermore, to achieve a more accurate determination of this efficiency, the Zn plating/stripping process in the Zn/V₂O₅ battery (with area capacity 0.7 mAh cm⁻²) was further simulated by using a more stringent condition in symmetrical Zn cells with Zn foils (thickness: 20 μm; 15.7 × excess) under galvanostatic condition (as shown below, current density 1 mA cm⁻², area capacity of 0.7 mAh cm⁻²). It is found that the symmetrical cell can indeed stabilize over 600 cycles, and the Zn foils remained in an integrated shape after cycling (**Reviewer 2 Fig. 1, shown below**).

In view of the above, it is hardly surprising that the theoretical 1.8 × excess Zn can support 600 cycles of the Zn/V₂O₅ cell.

Supplementary information, page21: insert: “Supplementary Notes for Fig. 19:

During the initial Zn plating process, Zn loss not only includes the side products formed by electrolyte decomposition, but also involves the associated side reactions between Zn and substrate surface²¹. As shown in Supplementary Fig. 19, the cell exhibits the highest CE of Zn plating/stripping when Ti using as the working electrode.”

Page4, line9: insert: “ZIBs with the V₂O₅ cathode accomplish a cyclability of 92.8% capacity retention over 800 cycles (99.9% CEs after activation), and have 97.5% capacity remaining after 600 cycles with an anode-limited design (theoretical 1.8 × excess Zn).”

Page10, line28: insert: “As an another reliable method^{16,57}, cyclic voltammetry (CV) was further applied to evaluate CE of ZES at an average deposition capacity ~0.61 mAh cm⁻² (Supplementary Fig. 21). Corresponding chronocoulometry curves (Fig. 2c) reveal that the

plating/stripping is highly reversible; the CE approaches 100% after the initial 30 conditioning cycles (an average CE of 99.7% for 200 cycles; see Supplementary Fig. 21). From this aspect, compared with other reported Zn electrolytes (see Supplementary Table 4)^{16,17,58}, ZES provides a more promising route for the realization of secondary Zn-metal cells to charge for hundreds of times, especially when the excess of Zn anode is limited.”

Page19, line20: insert: “In terms of energy density, it is acknowledged that the truly competitive ZIBs for industrial scenarios are achieved only if the Zn excess is limited at anode side¹⁶. Thus, we have attempted to estimate the utility of the ZES electrolyte on a more practical basis by a full cell with high-mass-loading V₂O₅ cathode (4.2 mAh cm⁻² based on the theoretical capacity of V₂O₅), and thin Zn foil anode with theoretical 1.8 × excess (11.7 mA h cm⁻²). A reversible capacity of 51 mAh g⁻¹ can be obtained after 600 cycles at an extremely high rate of 600 mA g⁻¹ (~13 minutes rate) with a capacity fading of only 0.0035% cycle⁻¹ (the capacity retention of 97.89%) (Fig. 6f).”

Fig. 2 Ionic transport and Zn plating/stripping behaviors in ZES. a) Current-time curves following DC polarization of the ZES at 0.005 V. Inset shows AC impedance spectra. b) Voltage profiles of galvanostatic Zn plating/stripping with the maximum oxidation potential of 0.5 V (vs. Zn/Zn²⁺) in ZES at a rate of 0.5 mA cm⁻² (1.0 mAh cm⁻²). The working and counter electrodes are Ti and Zn, respectively. c) Chronocoulometry curves of Zn plating/stripping in ZES based on cyclic voltammetry (CV) with Ti as the working electrode at a scan rate of 1 mV s⁻¹. Voltage responses of Zn/Zn symmetric cells d) in ZES and 1 M Zn(TFSI)₂ electrolytes at 0.1 mA cm⁻² (0.05 mAh cm⁻² for each half cycle) for 1,000 cycles (insets: the optical images of the cycled Zn after 180 cycles in 1 M Zn(TFSI)₂ (left) and 2,000 cycles in ZES (right)), and e) in ZES electrolyte (inset: in 1 M Zn(TFSI)₂ electrolyte) at 1 mA cm⁻² (0.5 mAh cm⁻² for each half cycle).

Supplementary Fig. 21 CV curves of Zn plating/stripping in ZES at a scan rate of 1 mV s^{-1} with a potential range of $-0.5 - 1.2 \text{ V}$ and an average deposition capacity of $\sim 0.61 \text{ mAh cm}^{-2}$. The working and counter electrodes are Ti and Zn, respectively. Inset: the variation of Zn plating/stripping CE with cycling.

Supplementary Fig. 19 (a) Voltage profiles and (b) corresponding CE of Zn plating/stripping on Ti, SS and Mo working electrodes (with same treatment condition), respectively, at 0.5 mA cm^{-2} (0.5 mAh cm^{-2}) in ZES.

Reviewer 2 Fig. 1 (a) Voltage responses of Zn/Zn symmetric cells in ZES at 1 mA cm^{-2} (0.7 mAh cm^{-2} for each half cycle) for 600 cycles. (b) The optical images of the cycled Zn after 600 cycles in ZES.

Comment 4: The baseline system, 1 M Zn(TFSI)_2 , triggers a different electrode chemistry in the formation of ZnO. The comparison with the DES system does not have a clear scientific rationale. It would be more appropriate to compare with a concentrated pure zinc solution and how the differences in ionic structure determines the cycling performance.

Our Response: These are very constructive suggestions.

1) Currently, there are still no well-recognized electrolytes to achieve reliable reversibility of Zn deposition/stripping both from electrochemical and thermodynamic aspects (*Nano Energy* **2019**, 57, 625). In view of this situation, recent works on exploring new Zn electrolytes use alkaline and neutral aqueous electrolytes for comparison, but several issues, in particular severe dendrite growth during cycling and metal passivation (*Nat. Mater.* **2018**, 17, 543; *J. Energy Storage* **2018**, 15, 304; *Chem. Commun.* **2018**, 54, 14097; *Adv. Energy Mater.* **2019**, 1900196) remain. In order to ensure that the anion and cation of control electrolyte group are consistent with those of the experimental group, Zn(TFSI)_2 is also applied in the present work. (please see the **response to Comment 7 of Reviewer 1**)

However, due to the high charge density of Zn^{2+} , Zn(TFSI)_2 is difficult to dissociate in common organic solvent systems, resulting in limited control over the coordination properties. Thus, in our work, water is used as the solvent since it can make Zn(TFSI)_2 more easily dissociated compared with acetonitrile or ionic liquids. Here, taking into account the pH and the ionic conductivity (**Supplementary Fig. 14**) of different concentrations, we screened 1 M Zn(TFSI)_2 as the control group (**Supplementary Fig. 14b**). Although the Zn(TFSI)_2 concentration can reach to 1.5 M based on water as the solvent, the stronger

acidity will bring about the corrosion of Zn anode and current collectors, and sustained consumption of water, resulting in poor long-term stability (*Angew. Chem. Int. Ed.* **2016**, 55, 2889; *Chem. Soc. Rev.* **2014**, 43, 5257; *Adv. Funct. Mater.* **2018**, 28, 1802564; *Nat. Mater.* **2018**, 17, 543).

2) Following the suggestions of reviewer, we also investigated the electrochemical properties of the concentrated pure Zn(TFSI)₂ solution (1.5 M Zn(TFSI)₂). The Zn salt will be precipitated in the 2 M Zn(TFSI)₂ after resting for 24 hours (**Supplementary Fig. 14a**). As shown in **Supplementary Fig. 15**, 1 M Zn(TFSI)₂ exhibits extended cycle life and lower polarization at both low (0.1 mA cm⁻²) and high (1 mA cm⁻²) rates compared with concentrated electrolyte (1.5 M Zn(TFSI)₂).

We further interrogated other high-concentration Zn electrolytes (30 m ZnCl₂, *Chem. Commun.*, 2018, 54, 14097, *Adv. Energy Mater.* **2019**, 9, 1900196; 4 M Zn(TfO)₂, *J. Am. Chem. Soc.*, **2016**, 138, 12894) that have been reported recently. As shown in **Reviewer 2 Fig. 2**, 1 M Zn(TFSI)₂ also exhibits more sustainable electrochemical cycling in contrast to those concentrated electrolytes at relatively low rate (0.1 mA cm⁻²). This significant difference may be ascribed to, in part, the more aggressive parasitic reactions in concentrated electrolytes because the acidity of the solution increases as the salts concentration increases (**Reviewer 2 Table 1**) (*Nat. Mater.* **2018**, 17, 543; *J. Am. Chem. Soc.* **2017**, 139, 18670).

3) As indicated by the reviewer, the differences in ionic structure determine the electrochemical performance. Taking into account the proposed anion-derived SEI formation, various Ace-based DESs have been prepared based on various common Zn salts (Zn(ClO₄)₂, Zn(CH₃COO)₂, Zn(BF₄)₂ and Zn(TFSI)₂), and thus the different ion species can be obtained. From the comparison in terms of the cycling performance for Zn plating/stripping and the oxidation stability (**Supplementary Fig. 17 and 36**), it is apparent that ZES indeed exhibits lower polarization and better stability compared with DESs based on other anions, demonstrating the TFSI⁻-induced SEI formation mechanism and the uniqueness of the SEI composition.

Recently, Wang et al. (*Nat. Mater.* **2018**, 17, 543) discovered that a mixture water-in-salt electrolyte comprising 1 m Zn(TFSI)₂ and 20 m LiTFSI (where m is molality (mol kg⁻¹)) delivers an extremely high CE of 99.7% on a platinum current collector. We also found that with the LiTFSI concentration increased, the electrochemical performance of the symmetrical Zn cell is greatly improved (**Reviewer 2 Fig. 3**). However, since this highly concentrated electrolyte is achieved by the sacrifice of introducing an excess of the lithium salt, several issues, like phase separation at low temperatures and high material cost, cannot be simply ignored.

For our ZES system, in addition to the new interphasial chemistry of anion-derived SEIs on Zn anode, the cost issue of electrolytes can be alleviated. Although Zn(TFSI)₂ is also used in the present ZESs, the raw material usage amount can be significantly reduced due to the major ingredient of cheap Ace (>80 mol.%). Second, given that DES features an intrinsic eutectic feature with rich intermolecular interaction, metal salts in DES have a higher thermodynamic stability (**Supplementary Fig. 2, 3**), as evidenced by the wide liquid range of ZESs. Moreover, the flexibilities of composition and ratio for ZES electrolytes

provide much room for improvement in ion-diffusion kinetics, interfacial reactions and even the cost control.

Supplementary information, page16: insert: “Supplementary Notes for Fig. 14:

In order to ensure the consistency between samples in terms of anion and cation, $\text{Zn}(\text{TFSI})_2$ is also applied in the control electrolyte group. However, due to the high charge density of Zn^{2+} , $\text{Zn}(\text{TFSI})_2$ is difficult to dissociate in organic solvents. Thus, water is used as the solvent since it can make $\text{Zn}(\text{TFSI})_2$ more easily dissociated, though the Zn salt will be precipitated in the 2 M $\text{Zn}(\text{TFSI})_2$ after resting for 24 hours, as shown in Supplementary Fig. 14a. Here, taking into account the pH, the ionic conductivity as well as the electrochemical performance (Supplementary Fig. 15) of different concentrations, we choose 1 M $\text{Zn}(\text{TFSI})_2$ as the control sample (Supplementary Fig. 14b).”

page17: insert: “Supplementary Notes for Fig. 15:

1 M $\text{Zn}(\text{TFSI})_2$ exhibits extended cycle life and lower polarization at both low (0.1 mA cm^{-2}) and high (1 mA cm^{-2}) rates compared with 1.5 M $\text{Zn}(\text{TFSI})_2$.”

Page17, line21: insert: “Note that ZES also exhibits lower polarization and better stability for Zn/Zn^{2+} reactions compared with DESs based on other Zn salts (Supplementary Fig. 36), demonstrating the TFSI^- -induced SEI formation mechanism and the uniqueness of the SEI composition.”

Supplementary Fig. 14 (a) Images of $\text{Zn}(\text{TFSI})_2$ aqueous electrolytes (resting for 10 hours). **(b)** Variation of pH and conductivity of $\text{Zn}(\text{TFSI})_2$ aqueous electrolytes with different concentrations.

Supplementary Fig. 15 Voltage responses of Zn/Zn symmetric cells in 1.5 M and 1 M Zn(TFSI)₂ electrolytes at (a) 0.1 mA cm⁻² (0.05 mAh cm⁻² for each half cycle) and (b) 1 mA cm⁻² (0.5 mAh cm⁻² for each half cycle), respectively.

Reviewer 2 Fig. 2 Voltage responses of Zn/Zn symmetric cells in 30 m ZnCl₂, 4 M Zn(TfO)₂, 1.5 M and 1 M Zn(TFSI)₂ electrolytes, respectively, at 0.1 mA cm⁻² (0.05 mAh cm⁻² for each half cycle).

Reviewer 2 Table 1

CEs of Zn plating/stripping of the reported concentrated Zn aqueous electrolytes.

ZnCl ₂		Zn(TfO) ₂		Zn(TFSI) ₂	
Molality (mol/kg)	pH	Molarity (mol/L)	pH	Molarity (mol/L)	pH
1	5.77	1	5.50	0.5	6.00
8	4.18	2	4.66	1	5.33
15	2.73	3	3.77	1.5	4.51
30	-0.28	4	2.92		

Reviewer 2 Fig. 3 Voltage responses of Zn/Zn symmetric cells in WiSE with varying LiTFSI concentrations at 0.02 mA cm^{-2} (0.5 h for each half cycle).

Supplementary Fig. 36 Voltage responses of Zn/Zn symmetric cells in DES electrolytes formed by different Zn salts ($\text{Zn}(\text{ClO}_4)_2$, $\text{Zn}(\text{CH}_3\text{COO})_2$ and $\text{Zn}(\text{TFSI})_2$) at 0.01 mA cm^{-2} (0.5 h for each half cycle).

In summary, the manuscript would make a good contribution to the field of zinc ion batteries by applying the SEI design method from other battery systems. It can be considered for publication if the concerns raised above are addressed.

Our Response: We appreciate the valuable evaluation of the reviewer and are grateful to the reviewer for the suggestions as to how we could improve our manuscript by clarifying particular points, which would provide some guidance for other multivalent metal-based batteries.

Attachment: List of changes in references of manuscript.

Previous manuscript	Revised manuscript
1-5	1-5
6	delete
N/A	6
7-25	7-25
N/A	26
26	28
27	27
28-31	29-32
32	delete
33-35	33-35
N/A	36-37
36-42	38-44
52	45
N/A	46
43	47
N/A	48-49
44-47	50-53
N/A	54
48	55
49	56
N/A	57-58
50-51	59-60
53-64	61-72
N/A	73

Reviewers' comments:

Reviewer #1 (Remarks to the Author):

Almost parts are appropriately revised according to reviewer's suggestions and therefore So the reviewer recommends Editor to publish it in this Journal.

Comment 5:

The reviewer recommends to the authors add impedance spectra and time dependence of dc current for symmetric cell for transference number measurements in Supporting information.

Masahiro SHIMIZU

Reviewer #2 (Remarks to the Author):

I appreciate the authors' effort in addressing the comments. I found the additional information with respect to the structural characterization to be helpful. Before recommending for acceptance, I would like the authors to address additional questions with respect to the data presented in Figure 6:

1) What is the difference between b and f? Both plots show full cell data at the same current density. The capacity in b is over 100 and the one in f is less than 50. I assume this is a test of the effect of loading levels?

2) In the reply to previous comments, the authors explained that the 1.8x excess is based on a full capacity of the oxide cathode and the actual excess is 15x. This information is critical in helping the readers to understand the scientific meaning of the data in Figure 6f and needs to be clearly explained in the main text. Presumably, the baseline 1M electrolyte cell should work reasonably well as well under the condition shown in Figure 6f. Given the large excess in actual zinc, I generally interpret the results in Figure 6 as an investigation on cathode effects since there is not a shortage of zinc at any time. In that case, I would advocate for better explanations for the performance improvement in b and d when compared to the control. If the point of Figure 6 is to show the benefit of a higher efficiency Zn anode, a true anode capacity limited cell would be the best to reach a firm conclusion.

3) I suggest actual current density values be used for figure b, d, and f.

Response to Reviewers

We would like to thank reviewers for their interest and time. The manuscript is revised according to their comments and suggestions.

Reviewer 1:

Almost parts are appropriately revised according to reviewer's suggestions and therefore So the reviewer recommends Editor to publish it in this Journal.

Our Response: We thank the reviewer for the encouraging comments.

Comment 5:

The reviewer recommends to the authors add impedance spectra and time dependence of dc current for symmetric cell for transference number measurements in Supporting information.

Our Response: Thanks for this helpful suggestion. The impedance spectra and time dependence of dc current for symmetric cell for transference number measurements have been added to the revised **Supplementary Fig. 18a** as suggested.

Supplementary Fig. 18 (a) Current-time curves following DC polarization of the ZES at 0.005 V. Inset shows AC impedance spectra. **(b)** Plot of current versus time^{-1/2} for ZES. Both working and counter electrodes are Zn. Potential Step: -0.2 V (vs. Zn/Zn^{2+}).

Reviewer 2:

I appreciate the authors' effort in addressing the comments. I found the additional information with respect to the structural characterization to be helpful. Before recommending for acceptance, I would like the authors to address additional questions with respect to the data presented in Figure 6:

Our Response: We are grateful to the reviewer for the positive comments and constructive suggestions as to how we could improve our manuscript.

Comment 1: What is the difference between b and f? Both plots show full cell data at the same current density. The capacity in b is over 100 and the one in f is less than 50. I assume this is a test of the effect of loading levels?

Our Response: The reviewer makes a good point.

1) Both plots of Fig b and f show cell data at the same gravimetric current density (based on the active cathode material). The main differences are the loading of V_2O_5 cathode and corresponding areal capacity. The cell in Fig. 6b is composed of a V_2O_5 loading of 1.6 mg cm^{-2} with a theoretical areal capacity of 0.5 mAh/cm^2 , while the cell in Fig. 6f has a high loading of 14.3 mg cm^{-2} with a much higher theoretical areal capacity of 4.2 mAh/cm^2 . It is well known that the loading of active materials has a crucial impact on the battery capacity. For instance, a higher-loading V_2O_5 tends to offer long path for ionic diffusion, giving rise to an increase in the diffusion impedance of Zn^{2+} (*Adv. Funct. Mater.* **2014**, 24, 44–52). As a consequence, an inevitable reduction of the specific capacity was observed, particularly in high rate testing rates. Thus, the capacity in Fig. 6b is over 100 mAh g^{-1} and the one in f is $\sim 50 \text{ mAh g}^{-1}$ at a same 600 mA/g . More importantly, a Zn foil with a thickness of $20 \mu\text{m}$ (14.28 mg cm^{-2} , 11.7 mAh cm^{-2} ; the mass ratio between Zn and V_2O_5 is 1:1) was applied for the cell in Fig. 6f. This cathode-anode configuration is apparently stricter than that of the Fig. 6b case (the mass ratio between Zn and V_2O_5 is 58:1) and also those of previous works on ZIBs.

2) We agree with the reviewer that Fig. 6b and 6f is a test of the effect of loading levels. Just as mentioned by the reviewer in **Comment 2** “*Given the large excess in actual zinc, the results in Figure 6 as an investigation on cathode effects since there is not a shortage of zinc at any time.*” We very appreciate the reviewer for spotting this oversight. To avoid misunderstanding, we have changed the description of Fig 6b and 6f in the revised manuscript, as shown below.

Page 4: rephrase: “With this in-situ anode protection, ZIBs paired with a V_2O_5 cathode accomplish a cyclability of 92.8% capacity retention over 800 cycles (99.9% CEs after activation), and were demonstrated to cycle up to 600 times along with a capacity fading of only $0.0035\% \text{ cycle}^{-1}$ under a practical cathode-anode coupling configuration (Zn: V_2O_5 mass ratio of 1:1; areal capacity of $> 0.7 \text{ mAh cm}^{-2}$).”

Page 19: rephrase: “Although cycling with a low areal capacity has been demonstrated to assist in maintaining a uniform morphology for metallic anodes⁷⁰,

material loadings must be rationally optimized to yield the truly competitive ZIBs for industrial scenarios¹⁶. Thus, we have attempted to estimate the utility of the ZES electrolyte on a more practical basis by a full cell with a high-mass-loading V_2O_5 cathode and a thin Zn foil (20 μm thickness, $\sim 11.7 \text{ mAh cm}^{-2}$). When the V_2O_5 loading is as high as 14.3 mg cm^{-2} , Zn//ZES// V_2O_5 cell still delivers a reversible capacity of 51 mAh g^{-1} (based on the mass of V_2O_5) after 600 cycles at an extremely high rate of 8.43 mA cm^{-2} with a capacity fading of only $0.0035\% \text{ cycle}^{-1}$ (the capacity retention of 97.89%) (Fig. 6f).”

Fig. 6 Electrochemical properties of ZIBs. **a)** Typical CV curves of the Zn/ V_2O_5 cell using ZES at a scan rate of 0.5 mV s^{-1} . **b)** Charge/discharge cycling performance and CE of the Zn/ V_2O_5 cells with ZES (after activation under 1 A g^{-1}) and 1 M Zn(TFSI)_2 electrolytes at 600 mA g^{-1} (0.79 mA cm^{-2}). **c)** Charge/discharge curves at various current densities in ZES. **d)** Rate performance of ZES and 1 M Zn(TFSI)_2 electrolytes. **e)** XRD patterns of the V_2O_5 cathode at different voltage states of the first cycle in ZES (10 mA g^{-1}). **f)** Long-term cycling performance of the Zn//ZES// V_2O_5 cell with the Zn: V_2O_5 mass ratio of 1:1 at 8.43 mA cm^{-2} (after activation under same rate; the capacity is calculated based on the total mass of cathode and anode). **g)** Typical galvanostatic charge/discharge profiles and CV curves (inset) of the Zn//ZES// Mo_6S_8 cell with ZES electrolyte. The current densities are calculated on the activated materials of cathode.

Comment 2: In the reply to previous comments, the authors explained that the 1.8x excess is based on a full capacity of the oxide cathode and the actual excess is 15x. This information is critical in helping the readers to understand the scientific meaning of the data in Figure 6f and needs to be clearly explained in the main text. Presumably, the baseline 1M electrolyte cell should work reasonably well as well under the

condition shown in Figure 6f. Given the large excess in actual zinc, I generally interpret the results in Figure 6 as an investigation on cathode effects since there is not a shortage of zinc at any time. In that case, I would advocate for better explanations for the performance improvement in b and d when compared to the control. If the point of Figure 6 is to show the benefit of a higher efficiency Zn anode, a true anode capacity limited cell would be the best to reach a firm conclusion.

Our Response: This is indeed a very critical suggestion to correct the controversial statement and illustration in our work.

1) Considering the practical application of ZIBs for industrial scenarios, we explored the performance of Zn/V₂O₅ full cell with a practical cathode-anode coupling configuration (the mass ratio between Zn and V₂O₅ is 1:1) in this work. Because the minimum thickness of available Zn foils is in the range of 10-20 μm, the loading cathode has to be increased. Thus a high V₂O₅ loading of 14.3 mg cm⁻² was applied, which significantly increases the areal capacity of the full cell (actual 0.7 mAh cm⁻² at 8.43 mA cm⁻²), but at the expense of specific gravimetric capacity (as mentioned in the **response to Comment 1**). This caused a greater difference between the theoretical and the actual excess especially at high charge/discharge rates. Although the deviation is obvious, the actual 15 × excess is of certain reference significance compared with the Zn sheet (over 100 × excess Zn) used in previous works on ZIBs (*J. Electrochem. Soc.* **2015**, 162, A1439; *J. Am. Chem. Soc.* **2017**, 139, 9775), which is in line with the practical application requirements. In sharp contrast, the capacity of the cell with the baseline 1 M Zn(TFSI)₂ rapidly decayed to 3 mAh g⁻¹ after only 60 cycles under a same condition in Fig. 6f, as shown below (**Response Fig. 1**).

Meanwhile, given the actual Zn excess, we also agree that the Fig. 6f is more appropriate as an investigation on cathode effects proposed by the reviewer. After all, the actual 15× Zn excess is not rigorous enough to explore the issue of anode excess. Therefore, we have made changes on description (the mass ratio between Zn and V₂O₅ was set to 1:1 in this work) in the corresponding part of the revised manuscript, as shown below.

2) The performance improvement in Fig. b and d when compared to the control is mainly due to uncontrolled dendrites (**Fig. 3, 5 and S20**) and notorious side-reactions (H₂ evolution, passivation and corrosion; the formation of water-insoluble ZnO and xZnCO₃·yZn(OH)₂·zH₂O, etc. is shown in **Fig. S25**) occurred at the Zn-electrolyte interface (*Adv. Energy Mater.* **2018**, 8, 1702097; *Nano Energy* **2019**, 57, 625), resulting in the inevitable decay of the specific capacity. In particular, the formation of insulating ZnO passivation on the surface of the Zn blocks a reversible Zn plating/stripping at anode, giving rise to an increase in Ohmic polarization of cells, and this is an important factor leading to the rapid decay of the battery capacity in 1 M Zn(TFSI)₂ (*Joule* **2018**, 3, 1; *J. Energy Storage* **2018**, 15, 304; *Nat. Chem.* **2018**, 10, 532). Based on this suggestive comment, our manuscript has been revised accordingly.

3) Furthermore, by further reducing the Zn:V₂O₅ mass ratio to 0.5:1 (thinner Zn

foil: 10 μm thickness, 7.14 mg cm^{-2}), a higher energy density of 40.9 Wh kg^{-1} at rate of 2.81 mA cm^{-2} was achieved (Supplementary Fig. 43). In the case of the development of Zn^{2+} -storage cathodes taking into account stability, capacity and operation voltage simultaneously (*Chem*, 2019, 5, 896), there is still vast scope for improvement in energy density of ZES-based ZIBs.

Page 4: rephrase: “With this in-situ anode protection, ZIBs paired with a V_2O_5 cathode accomplish a cyclability of 92.8% capacity retention over 800 cycles (99.9% CEs after activation), and were demonstrated to cycle up to 600 times along with a capacity fading of only 0.0035% cycle^{-1} under a practical cathode-anode coupling configuration (Zn: V_2O_5 mass ratio of 1:1; areal capacity of $> 0.7 \text{mAh cm}^{-2}$).”

Page 19: insert: “In sharp contrast, the capacity of the cell with 1 M Zn(TFSI)₂ rapidly decayed to 61.9 mAh g^{-1} (capacity retention $< 50\%$) after only 150 cycles, which is mainly ascribed to the formation of the insulating passivation layer on Zn anode (Supplementary Fig. 25) that blocks the Zn^{2+} interfacial transport, and the resulting increase in polarization^{10,23,26}.”

Page 19: rephrase: “Although cycling with a low areal capacity has been demonstrated to assist in maintaining a uniform morphology for metallic anodes⁷⁰, material loadings must be rationally optimized to yield the truly competitive ZIBs for industrial scenarios¹⁶. Thus, we have attempted to estimate the utility of the ZES electrolyte on a more practical basis by a full cell with a high-mass-loading V_2O_5 cathode and a thin Zn foil (20 μm thickness, $\sim 11.7 \text{mAh cm}^{-2}$). When the V_2O_5 loading is as high as 14.3 mg cm^{-2} , the Zn//ZES// V_2O_5 cell can be cycled still shows stable operation over 600 cycles at a high rate of 8.43 mA cm^{-2} with a capacity fading of only 0.0035% cycle^{-1} (the capacity retention of 97.89%) (Fig. 6f). In contrast to most of the previously reported ZIBs, wherein much excessive Zn needs to be used for prolonging the cycle life, the mass ratio between Zn and V_2O_5 was set to 1:1 in this cell. Based on the total mass of cathode and anode, the capacity is calculated to be 25.5 mAh g^{-1} , corresponding to an energy density of 25.8 Wh kg^{-1} . Additionally, further reducing the Zn: V_2O_5 mass ratio to 0.5:1 can provide an improved energy density of 40.9 Wh kg^{-1} (Supplementary Fig. 43). In the case of the development of the Zn^{2+} -storage cathodes taking into account stability, capacity and operation voltage simultaneously, there is still vast scope for improvements in energy density of ZES-based ZIBs⁷¹.”

Page 20: rephrase: “With this interface modulation, dendrite-free and intrinsically stable Zn plating/stripping can be realized at the areal capacity of $> 2.5 \text{mAh cm}^{-2}$ or even under a common dilute aqueous electrolyte system. Zn//ZES// V_2O_5 cells present remarkable electrochemical reversibility (an average CE of $\sim 99.9\%$, superior to most aqueous ZIBs^{9,71,72}) and laudable capacity retention even under rigorous but practically desirable cathode-anode loading conditions.”

Fig. 6 Electrochemical properties of ZIBs. a) Typical CV curves of the Zn/V₂O₅ cell using ZES at a scan rate of 0.5 mV s⁻¹. b) Charge/discharge cycling performance and CE of the Zn/V₂O₅ cells with ZES (after activation under 1 A g⁻¹) and 1 M Zn(TFSI)₂ electrolytes at 600 mA g⁻¹ (0.79 mA cm⁻²). c) Charge/discharge curves at various current densities in ZES. d) Rate performance of ZES and 1 M Zn(TFSI)₂ electrolytes. e) XRD patterns of the V₂O₅ cathode at different voltage states of the first cycle in ZES (10 mA g⁻¹). f) Long-term cycling performance of the Zn//ZES//V₂O₅ cell with the Zn:V₂O₅ mass ratio of 1:1 at 8.43 mA cm⁻² (after activation under same rate; the capacity is calculated based on the total mass of cathode and anode). g) Typical galvanostatic charge/discharge profiles and CV curves (inset) of the Zn/Mo₆S₈ cell with ZES electrolyte. The current densities are calculated on the activated materials of cathode.

Response Fig. 1 Long-term cycling performance of the Zn//1 M Zn(TFSI)₂//V₂O₅ cell with the Zn:V₂O₅ mass ratio of 1:1 at 8.43 mA cm⁻² (the capacity is calculated based on the (cathode + anode) mass).

Supplementary Fig. 43 The charge/discharge curves of Zn/V₂O₅ cell with ZES between 0.6 and 1.8 V at 2.81 mA cm⁻² (Zn:V₂O₅ mass ratio of 0.5:1, , after activation).

Comment 3: I suggest actual current density values be used for figure b, d, and f.

Our Response: We thank the reviewer for the constructive suggestion. We have changed the description of current density values in Fig 6b, 6d and 6f in the revised manuscript as suggested, as shown below.

Fig. 6 Electrochemical properties of ZIBs. a) Typical CV curves of the Zn/V₂O₅ cell using ZES at a scan rate of 0.5 mV s⁻¹. b) Charge/discharge cycling performance and CE of the Zn/V₂O₅ cells

with ZES (after activation under 1 A g^{-1}) and 1 M Zn(TFSI)_2 electrolytes at 600 mA g^{-1} (0.79 mA cm^{-2}). **c)** Charge/discharge curves at various current densities in ZES. **d)** Rate performance of ZES and 1 M Zn(TFSI)_2 electrolytes. **e)** XRD patterns of the V_2O_5 cathode at different voltage states of the first cycle in ZES (10 mA g^{-1}). **f)** Long-term cycling performance of the $\text{Zn//ZES//V}_2\text{O}_5$ cell with the $\text{Zn:V}_2\text{O}_5$ mass ratio of 1:1 at 8.43 mA cm^{-2} (after activation under same rate; the capacity is calculated based on the total mass of cathode and anode). **g)** Typical galvanostatic charge/discharge profiles and CV curves (inset) of the $\text{Zn/Mo}_6\text{S}_8$ cell with ZES electrolyte. The current densities are calculated on the activated materials of cathode.

REVIEWERS' COMMENTS:

Reviewer #2 (Remarks to the Author):

The reviewers comments have been addressed and the manuscript is suitable for publication.